# 3D mechanical characterization of single cells and small organisms using acoustic manipulation and force microscopy

Nino F. Läubli [1,5], Jan T. Burri [1,5], Julian Marquard[1], Hannes Vogler [2], Gabriella Mosca [2], Nadia Vertti-Quintero [3], Naveen Shamsudhin [1], Andrew deMello[3], Ueli Grossniklaus [2], Daniel Ahmed [1,4 ✉] & Bradley J. Nelson[1]

Quantitative micromechanical characterization of single cells and multicellular tissues or organisms is of fundamental importance to the study of cellular growth, morphogenesis, and cell-cell interactions. However, due to limited manipulation capabilities at the microscale, systems used for mechanical characterizations struggle to provide complete three-dimensional coverage of individual specimens. Here, we combine an acoustically driven manipulation device with a micro-force sensor to freely rotate biological samples and quantify mechanical properties at multiple regions of interest within a specimen. The versatility of this tool is demonstrated through the analysis of single *Lilium longiflorum* pollen grains, in combination with numerical simulations, and individual *Caenorhabditis elegan*s nematodes. It reveals local variations in apparent stiffness for single specimens, providing previously inaccessible information and datasets on mechanical properties that serve as the basis for biophysical modelling and allow deeper insights into the biomechanics of these living systems.

[1] Multi-Scale Robotics Lab, ETH Zurich, Zurich, Switzerland. [2] Department of Plant and Microbial Biology & Zurich-Basel Plant Science Center, University of Zurich, Zurich, Switzerland. [3] Institute for Chemical and Bioengineering, ETH Zurich, Vladimir-Prelog-Weg 1-5/10, Zürich, Switzerland. [4] Acoustic Robotics Systems Lab, ETH Zurich, Rüschlikon, Switzerland. [5]These authors contributed equally: Nino F. Läubli, Jan T. Burri. ✉email: dahmed@ethz.ch

The investigation of mechanical properties at the tissue, cellular, and subcellular levels is of fundamental importance for microbiology, cell biology, developmental biology, and medicine. A large number of methods, such as magnetic twisting cytometry, particle-tracking microrheology, optical stretching, or parallel-plate rheology have been applied to characterize mechanical attributes of single cells or cell layers[1]. Other popular approaches include indentation techniques such as atomic force microscopy (AFM) or cellular force microscopy (CFM)[2–6].

While broadly successful, all the above methods lack robust manipulation capabilities, which limits their ability to access all regions of a biological sample. Three-dimensional (3D) mechanical quantification of individual samples is restricted to larger length scales and requires manual handling of the specimen to obtain geometric information. The implication is that, at the microscale, significant assumptions and simplifications are necessary when modeling complex cellular systems or organisms. Additionally, the need to combine measurements on multiple individuals to ensure complete coverage of specimen structure strongly reduces experimental accuracy due to ensemble averaging. Diverse techniques based on magnetism, hydrodynamics, or optical fields have been used to successfully rotate a large number of objects at the microscale[7–10]. Unfortunately, most of these techniques rely on specific specimen properties, which compromises their applicability to biological samples of varying size. Acoustic waves have been widely used within microfluidic environments to perform functional operations, such as mixing, sorting, and drug delivery. Due to their high controllability and recognized biocompatible characteristics[11], acoustofluidic excitation has been used for particle and sample manipulation and rotation via surface acoustic waves[12], solid actuators[13], or trapped microbubbles in closed fluid chambers as well as open setups[14–20].

Here, we combine micro-indentation with an acoustically driven, bubble-based device for non-invasive trapping and 3D characterization of micron-sized objects. During the mechanical characterization process, a probe is pressed into the specimen with the resulting forces and indentation depths being recorded. The combination of 3D acoustic manipulation with micro-indentation enables the quantification of different surface regions of single specimens and, thus, expands the current possibilities of biological research at the microscale. First, we describe the structure and operation of our system and quantify its manipulation capabilities. Then, we present the advantages of our method by mechanically characterizing plant cell walls. This combination of techniques allows access to all surface regions of an individual cell, drastically reducing the influence of biological scattering on the results. We characterize the apparent stiffness ratio based on the apparent stiffness of different cell wall components of individual pollen grains in deionized water and examine our experimental observations through numerical simulations to disentangle the complex, intertwined contributions of geometry, heterogeneous material composition, and turgor pressure on the apparent stiffness obtained from mechanical characterizations. We then proceed by investigating the influence of the environment on the structural stability of the plant specimen through the quantification of pollen grains in water or calcium chloride ($CaCl_2$) solution and compare our observations to apparent stiffness values obtained for dehydrated specimens. Finally, we demonstrate the utility of our platform for measurements at the organism level by detecting the influence of internal organs on the mechanical properties of surrounding tissue in different regions of individual nematodes.

## Results

**3D manipulation and indentation.** The acoustic manipulation device enables precise and controlled rotation of a specimen and provides access to different regions of a sample (Fig. 1a–c and Supplementary Fig. 1). We designed an open-microchannel arrangement that contains linear arrays of rectangular microcavities. An acoustically-actuated microbubble is well suited for 3D manipulation because of its ability to focus acoustic energy and trap micron- to millimeter-sized specimens. The open-microchannel enables the force probe to directly interact with the specimen. Due to hydrophilic/hydrophobic interactions, the injected liquid causes air to be trapped within the predefined microcavities of the hydrophobic PDMS, leading to the formation of locally confined microbubbles. When the microbubbles are acoustically excited by the piezoelectric transducer, rapid liquid recirculation occurs near the interface. Such localized liquid recirculation or micro-vortices are also known as microstreaming, with the shape and strength controlled by the acoustic excitation frequency and the voltage applied to the piezo transducer. Depending on these input parameters, in-plane and out-of-plane micro-vortices can be formed in the surrounding liquid (Supplementary Fig. 2). Examples of such controlled liquid recirculations close to a microbubble can be visualized using fluorescent tracer particles, as shown in Supplementary Movies 1 and 2.

We integrated the manipulation device with the CFM setup to form an indentation-based mechanical characterization system[5]. As the specimen was delivered to the manipulation device, the linear array of the microbubbles attracted the specimen and trapped it. In contrast to closed microfluidic devices, the specimens can be directly positioned near the microbubbles, e.g., using pipettes, and no external pumps are required. The oscillation of the microbubble produced acoustic microstreaming and simultaneously exerted acoustic radiation forces on the specimen. The radiation force, generated by a microbubble, scales with the sample volume[21]. Such a scaling relationship is particularly attractive since specimens between the micro- and millimeter scale, such as *L. longiflorum* (lily) pollen grains and *C. elegans* nematodes, can be trapped effectively. The actuated microbubble traps the specimen, and microstreaming rotates the specimen. The linear array of the microbubbles and the modulation of the acoustic excitation signals ensure stable trapping and a controlled rotation of the specimen along its long-axis (see Fig. 1). Importantly, since the system operates at low Reynolds numbers (see Supplementary Note 1), rotation of a specimen stops immediately and without drift when the acoustic signal is terminated. This allows for precise positioning of the force sensor at a user-defined region on the exterior surface of the specimen to perform the mechanical characterization. As the rotational velocity can vary slightly depending on the specimen as well as the microbubble, the rotation of the sample was stopped manually. We performed mechanical tests based on micro-indentation of both soft and hard living samples using a commercially available microelectromechanical systems (MEMS)-based capacitive force sensor. A 3D positioner navigates the sensor probe close to the specimen and positions it in the field of view before fine positioning is performed via the microscope stage that moves the manipulation device and the sample. Vertical indentations of the exterior surface of the specimen is then performed in z-direction using a high-precision piezo stage. Figure 1b shows the deformation of the specimen by the force probe. Supplementary Movie 3 demonstrates the manipulation and indentation procedure. The CFM system (Fig. 1d) directly measures the applied force, $F$, and the corresponding displacement, $z$, with the apparent stiffness, $k$, of the sample being defined as the slope of the force-displacement curve ($\Delta F/\Delta z$). The apparent stiffness contains local information on various specimen components and thus is, e.g., in plant cells, influenced by the mechanical properties of the cell wall, the internal turgor pressure, as well as geometric factors. The technique shows a

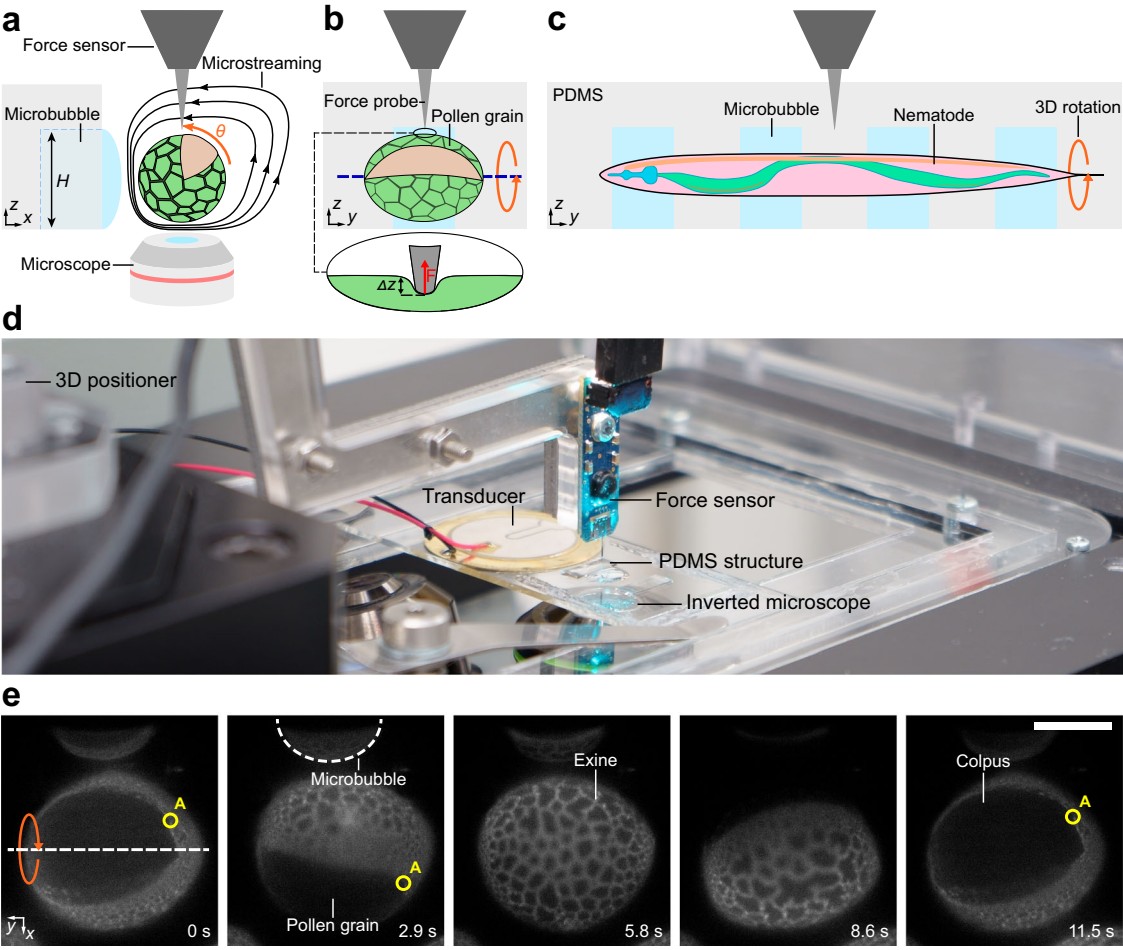

**Fig. 1 Setup for acoustic manipulation and micro-indentation. a**, **b** Side and front view schematics describing the 3D characterization of a pollen grain. Acoustic streaming, generated by exciting trapped microbubbles, allows the trapping of pollen grains and their re-orientation with respect to the force sensor. This enables the 3D mechanical characterization of a sample. $H$ denotes the height of the microbubble, $\theta$ the rotational angle of the specimen, and $x$, $y$, $z$ define the coordinate system where the force $F$ applied by the sensor in $z$-direction leads to a displacement $\Delta z$. **c** A schematic showing the manipulation and indentation of a single *C. elegans* nematode. By exciting multiple parallel microbubbles, the same design used for pollen grains can be applied to manipulate a nematode. **d** A photograph of the setup with the manipulation device, consisting of a PDMS structure and a piezoelectric transducer, as well as the force sensor used for mechanical characterization of the specimens. **e** A slowly rotating auto-fluorescent pollen grain. Stable rotation of the pollen grain is controlled via acoustic excitation. A denotes a specific location on the pollen grain's surface which is tracked during the rotation of the specimen. Scale bar: 50 μm.

high repeatability for the mechanical characterization of biological specimens with an average coefficient of variation of 4.8% ($n = 5$, $m = 50$, see Supplementary Fig. 1b and Supplementary Note 2). Please note that the derived reproducibility can also be affected by local changes in the biological specimen induced through repeated indentations[22].

**3D mechanical characterization of plant cells.** A fundamental process of plant reproduction is pollination, where a dehydrated pollen grain (Fig. 2a) is transferred from the anther to the carpel. Once a pollen grain lands on the stigma, the receptive part of the carpel, it rapidly rehydrates and unfolds (Fig. 2b)[23]. A cell wall, which is made of a complex composite material, surrounds and protects the cellular contents of the pollen grain. The cell wall consists of a resistant outer wall, the exine, and an inner wall, the intine (Fig. 2c, d). While most of the surface of the lily pollen grain is covered with both exine and intine, the exine is missing at the colpus. After rehydration, the pollen grain germinates and the pollen tube emerges at the colpus, where the intine is exposed. The pollen tube then grows towards the ovule and delivers the two sperm cells, which it carries as internal cargo, to effect double

fertilization. The colpus exhibits a softer material than the stiffer exine, which largely consists of the toughest known biopolymer, i.e., sporopollenin[24,25]. However, very little is known about the correlation between the mechanical properties of the different surface regions, and no complete mechanical characterization of an individual pollen grain has been reported. Additionally, the changes in stiffness caused by the transition between the folded (dehydrated) and unfolded (hydrated) states have yet to be measured.

We used the 3D acoustic manipulation device combined with CFM to quantify the apparent stiffness of a hydrated pollen grain (major axis of $128.5 \pm 9.9$ μm; minor axis of $98.3 \pm 5.8$ μm) at different surface regions. Hydrated lily pollen grains were submerged in deionized water and the force probe was positioned directly above the sample to indent the cell wall. Force-displacement data were recorded to calculate the apparent stiffness (Fig. 2f). When the piezo transducer was actuated, the sample rotates along the $y$-axis. After the specimen reached a new angular position, the acoustic signal was turned off and a new indentation measurement was performed. The indentation procedure was repeated 10 times for each pollen grain at various

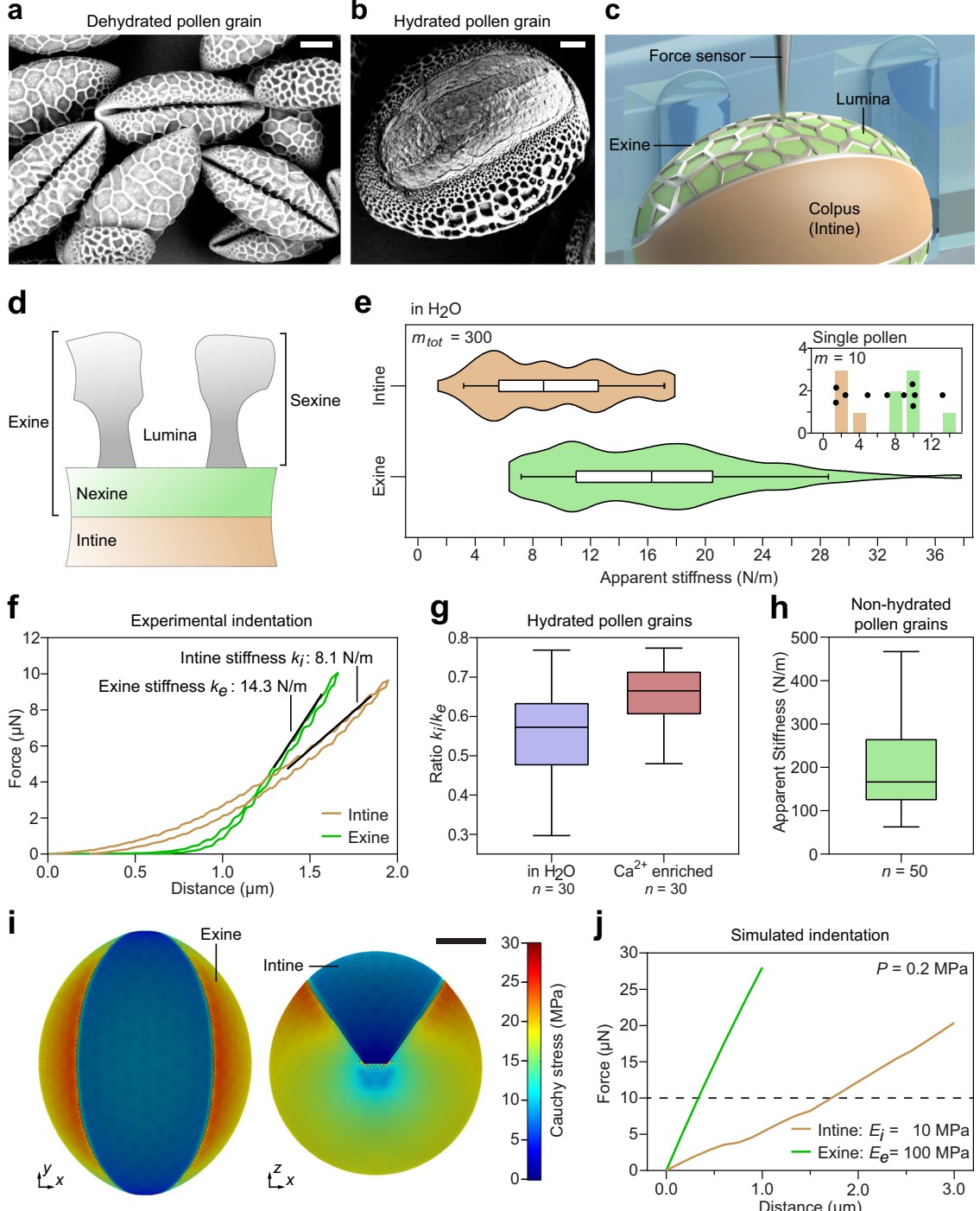

orientation angles to ensure full coverage of the surface. We obtained apparent stiffness measurements ($m = 300$ indentations) from $n = 30$ pollen grains. The data suggest a difference in material stiffness of the intine and exine regions. Figure 2e shows the distribution of all apparent stiffness values for intine (brown) and exine (green) material. The mean intine and exine apparent stiffness values were $9.3 \pm 4.4$ N/m and $16.5 \pm 6.6$ N/m, respectively. The inset shows 10 measurements for a single pollen grain, clearly exhibiting the increased stiffness of the exine compared to the intine.

The observed large value range over the 30 samples can be explained by structural differences between the pollen grains. Such variations could not be observed by optical examination, as

no correlation between the measured apparent stiffness and shape or size of the pollen grain was found (Supplementary Fig. 3). Therefore, we evaluated the mechanical properties of pollen grains individually. To combine the different measurements ($n = 10$) from a single pollen, the determined apparent stiffness values $k$ were assigned to the soft colpus (intine, $k_i$) or the stiff exine ($k_e$) regions (Fig. 2f). For each pollen grain, a single ratio $k_i/k_e$ is calculated, i.e., the mean colpus apparent stiffness of a single pollen divided by the mean exine apparent stiffness of the same pollen. This leads to a strong reduction in variance and minimizes the influence of biological scatter. Although the apparent stiffness values of the pollen grains differ vastly, the calculated ratios are similar. Such an analysis highlights the

**Fig. 2 3D indentation of lily pollen grains. a** A scanning electron microscopy (SEM) image of dehydrated lily pollen grains; note that the intine is not accessible. **b** An SEM image of a fully hydrated lily pollen grain, exposing the intine at the colpus. **c** A schematic of a lily pollen grain highlighting the different surface areas, i.e., the soft colpus and the stiff exine with the lumina. **d** A graphic showing the structure of a pollen grain cell wall in cross-section. At the colpus, the exine is absent and the intine is exposed to the surroundings. **e** Two violin plots containing all measurements ($m = 300$) for $n = 30$ biologically independent lily pollen grains in deionized water. The data is divided into intine (brown) and exine (green) stiffness values. The box represents the interquartile range, the center line represents the median, and the whiskers represent the 5th and 95th percentiles. The maxima and minima are denoted by the start and end of the violin plots. The inset shows the independent measurements ($m = 10$) for a single pollen grain, where intine and exine are separated. The individual measurements are represented by dots. **f** A graph showing two indentation curves from different regions of the same pollen grain. The stiffness values $k_i$ and $k_e$ are obtained from the slopes. **g** Two boxplots to compare the stiffness ratios $k_i/k_e$ of lily pollen grains in deionized water (violet) to the ratios of lily pollen grains in $CaCl_2$ solution (red). The boxes represent the interquartile ranges, the center lines represent medians, and the whiskers denote the ranges of minima and maxima. The stiffness ratio of the pollen grains in a $CaCl_2$ solution significantly increased, indicating specific stiffening of the intine due to $Ca^{2+}$-mediated crosslinking of pectin. Each stiffness ratio was derived from $m = 10$ independent measurements on a single specimen. Each stiffness ratio distribution was calculated from $n = 30$ biologically independent samples. **h** The apparent stiffness distribution for the exine structure (green) of non-hydrated pollen grains. The data was collected through ($m = 50$) single indentations of 50 biologically independent samples. The box represents the interquartile range, the center line represents the median, and the whiskers denote the range of minima and maxima. **i** The pressurized (turgor pressure $P = 0.2$ MPa) simulation of an average lily pollen grain. The two cell wall layers, i.e., the intine and exine, are clearly visible with the intine being exposed at the colpus. The color bar shows Cauchy stress in the cell wall in MPa. The side view shows the slight variation in pollen grain curvature between different regions of the exine as well as at the colpus and has been taken into account for the simulated indentations. **j** Simulated indentations of the exine and intine resulting in apparent stiffness values $k_e$ and $k_i$ of 12.7 N/m and 7.9 N/m, respectively, for a material stiffness ratio $E_e/E_i$ of 10. The turgor pressure is $P = 0.2$ MPa and the Poisson's ratio is 0.3. The dashed line denotes the maximum force applied during experimental characterization. Scale bars: $a = 25$ μm, $b = 10$ μm, $i = 25$ μm. Source Data is available as a source data file for **e–h**, **j**.

importance of characterizing all surface regions of single pollen grains individually, since combining data from multiple specimens is likely to lead to a loss of information or misinterpretation. Figure 2g shows a boxplot (violet) presenting the $k_i/k_e$ apparent stiffness ratio distribution for hydrated lily pollen grains. The mean apparent stiffness ratio (and standard deviation) for lily pollen grains in deionized water is $0.56 \pm 0.1$.

It is worth noting that the apparent stiffness distributions for intine as well as exine measurements were found to be significantly different from a normal distribution (see Supplementary Note 3). However, no correlation was found with the force sensor used, the date of the experiment, the time of the measurement, the flower from which the pollen was derived, or the geometry of the pollen grain.

**Numerical simulation of plant cell indentation.** The apparent stiffness obtained from experimental indentation is, as shown in previous work[26–30], strongly affected by various parameters, such as local curvature, cell size, material properties, or turgor pressure. In order to disentangle such contributions and connect the apparent stiffness to the cell wall's elastic (Young's) modulus $E$ and turgor pressure $P$, we developed a FEM-based simulation of an average pollen grain (see Methods section). This simulation allowed us to assign various homogeneous material properties to the different cell wall layers as well as varying turgor pressures (see Fig. 2i and Supplementary Fig. 4a, b). The initial material stiffness for the exine was set to twice the Young's modulus of the intine. Due to the strong variability of the measured apparent stiffness among pollen grains, we decided to compare the resulting simulated values with the experimental measurements shown in Fig. 2f.

In a first step, the intine layer was indented at the center of the colpus (see Supplementary Fig. 4c). The resulting indentation curves are presented in Supplementary Fig. 4d and the computed apparent stiffness values are reported in Supplementary Table 1. By comparing the simulated data with the experimental observations, we found that, among the tested parameters, the best fitting combination of turgor pressure $P$ and the intine's Young's modulus $E_i$ was 0.2 MPa and 10 MPa, respectively. These values led to an apparent stiffness $k_i$ of 8.5 N/m (experimental $k_i = 8.1$ N/m; relative difference 4.8%). It is worth noting that, in selecting the best fitting parameters, not only the apparent

stiffness but also the absolute reaction force at 1.8 μm indentation depth, i.e., 10 μN, was considered.

We then performed indentation simulations for the exine layer using the same set of parameters. For the exine, the indentation location on the pollen grain was chosen opposite to the indentation site for the intine (Supplementary Fig. 4c). The resulting indentation curves are presented in Supplementary Fig. 4e and the corresponding apparent stiffness values can be found in Supplementary Table 1. The results of the simulation illustrate that the apparent stiffness of intine and exine do not follow the same relation as the corresponding Young's moduli, i.e., with the material stiffness of the exine being twice that of the intine, the apparent stiffness of the exine does not double as compared to the intine. Furthermore, for certain combinations of parameters, the apparent stiffness of the exine becomes even lower than that of the intine. For instance, with the best fitting combination for the intine ($P = 0.2$ MPa, $E_i = 10$ MPa), the apparent stiffness of the exine is not doubled but is, with 7.9 N/m, even lower than that of the intine with 8.5 N/m.

Therefore, we proceeded by increasing the ratio of the elastic moduli ($E_e/E_i$) from 2 to 10. When arriving at a ratio of 10, i.e., $E_e = 100$ MPa and $E_i = 10$ MPa, the exine's apparent stiffness (see Supplementary Table 1) reaches a value of 12.7 N/m (experimental $k_e = 14.3$ N/m; relative difference 11.8%), indicating that a material stiffness ratio $E_e/E_i$ of ~10 corresponds to the experimentally determined apparent stiffness ratio. The simulated indentation curves are presented in Fig. 2j. To quantify the effect of varying exine material properties onto the intine measurements, we simulated the indentation of the colpus again. Supplementary Figure 4f shows the corresponding indentation curves for material stiffness ratios of 2 and 10. While changes in exine material stiffness display only a small effect on the simulated properties of the intine, the resulting apparent stiffness value, i.e., $k_i = 7.9$ N/m, moved closer to our experimental measurements.

It is important to highlight that the simulated indentation curves are not perfectly linear but follow a qualitative sublinear trend, which deviates from the exponential curves of experimental characterizations. While this effect is less pronounced at very low values for turgor pressures $P$ and elastic moduli $E$ (see Supplementary Fig. 4d), the qualitative difference is indeed present for all tested parameters. To explore the source of this

behavior, an additional model was developed. Supplementary Fig. 4g shows the indentation curves for a spherical model with varying shell thicknesses (see Methods section). The observed shift from sublinear to superlinear in the first part of the indentation curves indicates that only fractions of the whole cell wall (for both intine and exine) are mechanically active, a prediction that we can currently not test, however.

**Environmental dependence of cell wall stiffness.** Calcium ($Ca^{2+}$) is an important ion for signaling as well as for physiological and biochemical processes during pollen germination and pollen tube growth[31,32]. Among other functions, it impacts the mechanical stability of plant cell walls. For example, $Ca^{2+}$ ions lead to crosslinking and stiffening of pectin in the cell wall during the rapid growth of pollen tubes[33]. Therefore, we investigated the influence of $Ca^{2+}$ on the mechanical properties of the pollen grains. The soft colpus contains pectin, which is absent in the hard exine structure[34]. Accordingly, the addition of $Ca^{2+}$ to the liquid medium is expected to only affect the stiffness of the colpus by complexation of $Ca^{2+}$ ions with de-esterified pectins, whilst the exine should remain unaltered[33,35]. However, given the large biological variation between grains observed previously, a full characterization of individual pollen grains is required to experimentally quantify mechanical changes. We administered lily pollen grains in 5 mM $CaCl_2$ into the manipulation device and measured their apparent stiffness (Supplementary Fig. 3). Higher stiffness ratios ($n = 30$) were observed compared to those obtained in water (Fig. 2g). This corresponds to the definition of the apparent stiffness ratio, i.e., the experimentally determined stiffness value of the colpus divided by the experimentally determined stiffness value of the exine, as only the intine rigidifies while the exine remains essentially unaffected by the $Ca^{2+}$ ions. Given the experimental data, it can be seen that $CaCl_2$ leads to an apparent stiffness for the intine which is on average 0.66 times the apparent stiffness value of the exine. Given the bimodal trend of the apparent stiffness, the stiffness ratios have been tested for normality (see Supplementary Note 3) and the difference of the stiffness ratios obtained for pollen in deionized water and $CaCl_2$ solution has been found significant ($p = 0.000312$) in subsequent statistical evaluations (see Supplementary Note 4). Therefore, our findings support the theory of partial cell wall stiffening through $Ca^{2+}$ ions.

Finally, we investigated the changes in apparent stiffness caused by hydration of the pollen grain. We performed 50 single point measurements on dry, dehydrated lily pollen grains. As dry, dehydrated pollen grains only expose the rigid exine[23], no acoustic manipulation was required. The detected average stiffness of 194 N/m (Fig. 2h and Supplementary Fig. 3c) is at least five times higher than the maximum stiffness measured for hydrated pollen grains. This suggests that the hydration phase causes significant changes in the cell wall matrix, e.g., through swelling. However, additional factors, such as solid intracellular features or more stable configurations of the cell wall cannot be excluded. Furthermore, it is important to highlight that the data obtained for dehydrated pollen grains follows a nearly non-normal distribution ($p = 0.0578$). Therefore, further investigations would be required to quantify the precise effect and allow for the examination of additional parameters, such as the influence of geometry, exine stability, and intracellular components.

**3D mechanical characterization of a multicellular organism.** The nematode *C. elegans* is a widely used model organism for the investigation of fundamental biological processes, which are often conserved in higher organisms through evolution[36,37]. In recent years, microfluidic devices have been used to immobilize and characterize mechanical properties of *C. elegans*[38,39]. In addition, acoustofluidic systems have been used to study nematodes[40], as well as for the precise manipulation of these highly motile specimens[12,41]. Such approaches are efficient yet gentle when manipulating *C. elegans* and, despite relying on paralysing drugs, have been shown to better preserve animal health and viability when compared to conventional, manual techniques that commonly involve destructive physical constraints such as irreversible glue. This aspect is especially crucial as it might limit the viability of the specimens and correspondingly our capabilities for long-term biomedical studies, such as research on neurodegenerative diseases for which the worm is preferably studied throughout its life-cycle[42,43].

As, unfortunately, many acoustofluidic devices rely on closed microchannels to combine acoustic excitation with hydrodynamic flows, direct access to the constrained animals is strongly limited. Conversely, experimental platforms that allow direct interactions with external tools often lack the ability to manipulate the nematode and access different regions of the body[6,44–48]. Consequently, data acquired from a single region are often generalized, which can lead to inaccurate conclusions or highly simplified models[46,49]. To address such drawbacks, we used our experimental platform for the analysis of multiple regions on individual nematodes in a biocompatible manner.

Paralyzed *C. elegans* (see Methods section) were trapped and oriented using acoustic radiation forces and acoustic streaming generated by the acoustically excited microbubbles (Fig. 3a). Figure 3b presents images from a complete out-of-plane rotation of a fluorescently labeled *C. elegans* worm (Supplementary Movie 4). The mechanical properties of the region of interest (Fig. 3a, blue rectangle) were measured via single point indentations. Lines of up to eleven measurement points, each separated by 20 μm along the longitudinal axis of the animal, were probed (Fig. 3c). Application of a maximum force of 3 μN (resulting in a maximum indentation depth of 10 μm) was sufficient to quantify the stiffness of the cuticle and underlying tissues, whilst low enough to prevent damage to the specimen. After each line of measurements, the specimen was acoustically rotated to allow for the indentation of previously inaccessible areas. Importantly, the measured apparent stiffness values are within the range of experimental results published previously[44].

Three-dimensional mechanical characterization of a single *C. elegans* nematode revealed large variations in apparent stiffness with alternating softer ($k_1 = 0.53 \pm 0.07$ N/m, $n = 20$) and stiffer ($k_2 = 0.75 \pm 0.11$ N/m, $n = 30$) areas (Fig. 3e, f). The stiffness values associated with the softer and stiffer areas were found to be significantly different ($p = 3.14e-10$) through statistical evaluations (see Supplementary Note 5). Based on the arrangement of the body wall muscles in nematodes as well as the ~90° separation of the detected varying stiffness values (see Fig. 3d and green areas in Fig. 3f), the observed difference in apparent stiffness might be caused by the body wall muscles. We think that this variation in stiffness is enhanced by the use of the paralysing drug[50], which increases the muscle tone when used in nematodes. Therefore, the softer regions would be directly above the eggs/embryos inside the gonad (where the dorsal muscles are not present), while stiffer measurements would be recorded in the region of the body wall muscles. However, it is important to note that the observed variations might also be caused by local variations in the cuticle or by complex interactions between various factors. Further investigations will be required to differentiate between these interpretations as well as to quantify the precise impact of drug-induced short-term paralyzation onto internal tissues and subsequent micro-indentations. Nevertheless, given these large differences, it is clear that the ability to quantify

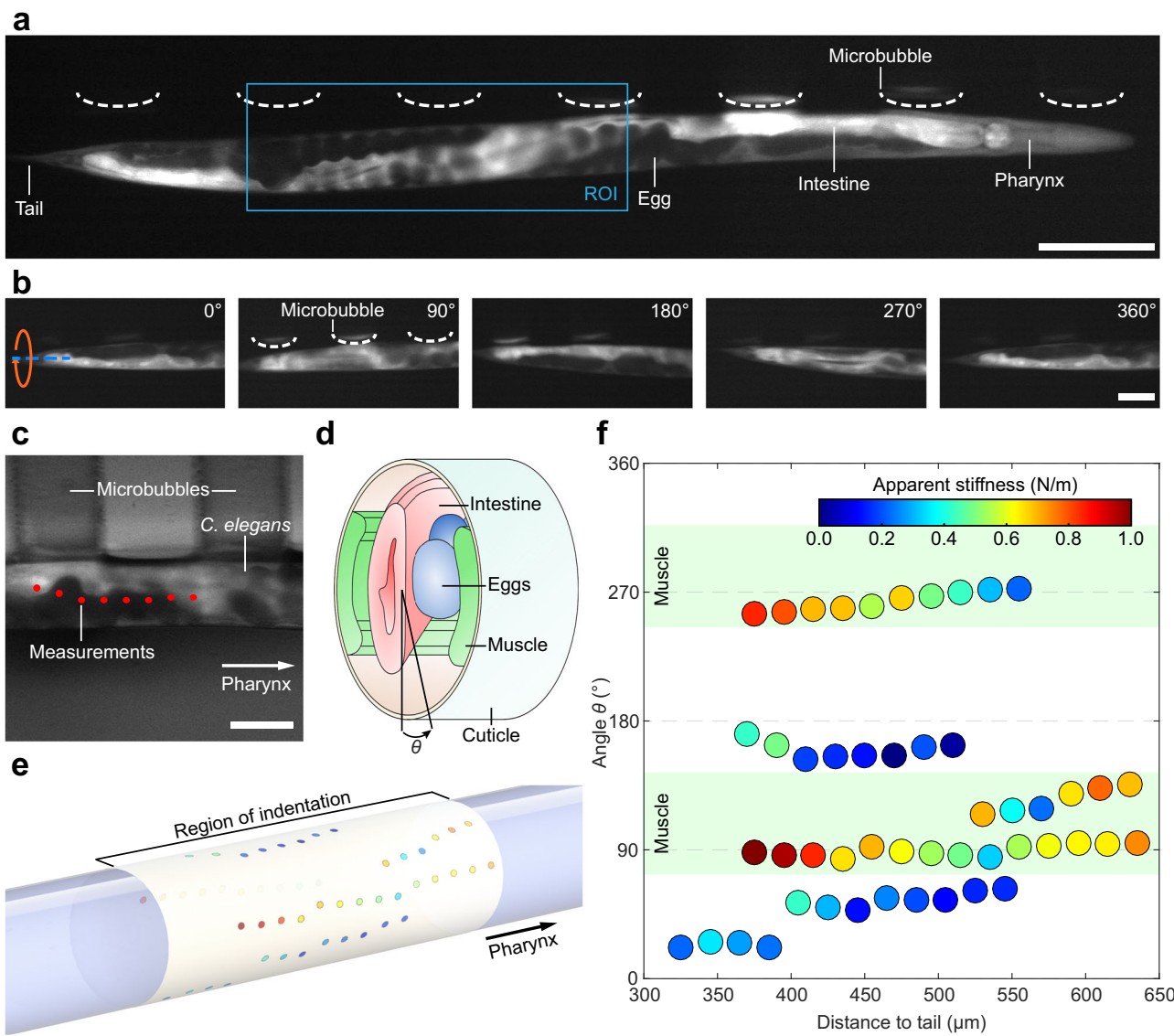

**Fig. 3 3D indentation of *C. elegans*. a** A fluorescence image of a *C. elegans* nematode trapped in the vicinity of microbubbles using acoustic radiation forces. The region of interest (ROI), where the mechanical characterization was performed, is highlighted in blue. **b** An image sequence showing different orientations of a *C. elegans* nematode. The specimen is rotated using acoustic streaming. **c** A stationary *C. elegans* worm near the microbubbles. Red dots indicate the individual positions of the micro-indenter probe during mechanical characterization. **d** Schematic illustrating the composition of internal organs which might lead to different stiffness values measured by micro-indentation and which can be used to extract the right orientation via fluorescence imaging. The angle θ denotes the location of the indentation in a cylindrical coordinate system. **e** 3D projection of the stiffness values onto the geometry of a *C. elegans* nematode showing the different positions around the characterized specimen. **f** The unfolded stiffness map with individual indentation positions. Multiple bands with different stiffness ranges can be observed. All presented measurements were performed on a single nematode to avoid noise caused by biological variation between specimens. For multicellular organisms, the stiffness does not necessarily quantify a single biological material, but may include the mechanical properties of several overlapping layers and, hence, should be treated as apparent stiffness. Green bands indicate the possible influence of muscular tissue. Scale bars: a = 100 μm, b = 50 μm, c = 50 μm. Source Data is available as a source data file for **f**.

the entire surface of individual nematodes is key, since tissues below the cuticle in single adult worms may have a strong influence on locally measured mechanical properties. Indeed, generalizations from 1D mechanical characterizations of fully developed adult nematodes can lead to errors and misinterpretations in the study of particular biological phenomena.

## Discussion

3D mechanical characterization at the micron to millimeter scale is crucial for a better understanding of intercellular interactions or the development and structural stability of cells, tissues, and organisms. In the current study, we present a instrumental platform that combines a microbubble-based manipulation device with CFM and allows for the 3D mechanical characterization of cells and organisms. The fabricated device enables the investigation of samples via microstreaming induced by oscillating microbubbles. Compared to existing acoustofluidic methods, microbubbles are able to generate strong vortices in the surrounding fluid while requiring low power inputs. Additionally, the more prominent out-of-plane streaming associated with an open-microchannel design simplifies the trapping and manipulation of the specimen. It should be noted that the temporal stability of the microbubbles is limited (i.e., they shrink over

time), and thus the rotational velocity of specimens will vary over time. However, this drawback is non-problematic since our design allows rapid cleaning and reuse. Accordingly, by drying and refilling the acoustofluidic chamber, new microbubbles are trapped and fresh specimen provided. Moreover, the temporal stability could be improved by increasing the length of the microcavities, and is the subject of further research and development. Additionally, high controllability over the produced microstreaming is essential to prevent damage to the fragile MEMS force sensor by inducing vibrations and to avoid attractive radiation forces between its sharp tip and the investigated specimen, which could alter the sample's orientation.

The advantages of our system have been demonstrated using plant and animal specimens. The characterization of individual lily pollen grains through the indentation of the cell wall demonstrated unreported correlations between intine and exine. It is worth noting that the bimodal behavior observed for the apparent stiffness of the intine as well as the exine has been eliminated through the derivation of the individual stiffness ratios. While substantial experimental and numerical investigations might be required to determine the cause of this bimodality, we propose that it might arise because pollen with a higher exine stiffness also expose higher intine values to form mechanically stable configurations, and vice versa. However, another possible explanation might be differences in hydration, as unfolded pollen do not have to be fully pressurized and a resulting difference in turgor pressure would simultaneously affect intine and exine measurements[51]. We then developed a numerical model to investigate the complex relationship between the measured apparent stiffness and different specimen properties, such as turgor pressure $P$ and Young's modulus $E$, and estimated the appropriate material stiffness ratio for the different cell wall layers. However, given the required simplifications and because the exine was represented as a homogeneous material, such estimations can only provide an indication for the average material stiffness. Nevertheless, our examination of this complex matter highlights that a certain measured apparent stiffness only partially reflects the effective material stiffness and opens a direction of research regarding mechanically active portions of cell walls. We proceeded by experimentally quantifying the influence of $Ca^{2+}$ ions on the apparent stiffness of the cell wall and observed significant variations induced by the liquid environment, although their exact source, e.g., changes in material stiffness, turgor pressure, or pollen grain geometry, remains to be determined. Lastly, by characterizing the exine structure of non-hydrated pollen grains, we revealed that their apparent stiffness values decrease upon hydration while the complex relationship between intracellular features, exine material stiffness, as well as cell wall configuration and curvature remains unclear. The measured apparent stiffness of non-hydrated pollen grains corresponds to the previously reported values for sporopollenin[25].

Finally, the use of parallel microbubbles allowed a controlled rotation and 3D mechanical characterization of *C. elegans* nematodes. Different regions displaying significant variations in stiffness were detected within individual specimen. Heterogeneity in measured mechanical properties might be the result of the underlying tissues, such as body wall muscles, over which the apparent stiffness was measured. However, other factors, such as potential local variations in the cuticle or a combination of multiple effects, cannot be neglected and further investigations will be required to differentiate between these interpretations. Furthermore, it is important to highlight that the detected variations might be enhanced through the application of chemical paralyzation and its precise impact has yet to be quantified. We expect that future experiments will focus on an assessment of the changes of stiffness in the body of *C. elegans* occurring through its developmental, reproductive, and post-reproductive stages, focusing on skin properties as well as organogenesis. Our results highlight the importance of manipulation capabilities during mechanical characterization to prevent misinterpretations or simplifications in modeling. Additionally, the applications could be expanded to the investigation of spheroids of transformed cells, with a view to quantifying mechanical alterations in cancer cells. Moreover, improvements in the imaging system will provide additional views (e.g., side views[52]) during indentation that will dramatically simplify subsequent analyses.

## Methods

**Cellular force microscopy**. The system was adapted from the previously reported CFM setup[5,53,54] and integrated into an inverted microscope for high-resolution bright field and fluorescent imaging. The system utilizes a commercially available MEMS-based capacitive force sensor from FemtoTools AG to measure forces during micro-indentation. Sensors with different force ranges are available and can be chosen according to the stiffness of the specimen (e.g., FT-S10'000 for dry pollen grains, or FT-S100 for hydrated pollen grains and *C. elegans*). A micro-indenter, a tungsten tip with a diameter <2 μm (T-4-22, Picoprobe R by GGB Industries INC), is attached to the sensor. The force sensor is connected to a 3D positioner (SLC-2475-S; SmarAct GmbH) with a 10-cm travel range and a closed-loop resolution of 50 nm, which is used to position the micro-indenter inside the field of view as well as close to the specimen prior to the experiment. The sensor keeps its relative position with respect to the microscope even when the microscope stage (containing the acoustic manipulation setup with the specimens) is moved. The subsystem connected to the sensor is placed onto a piezo flexure-guided nanomanipulation system (P-563.3CD PIMars; Physik Instrumente (PI) GmbH & Co), which is used to perform the micro-indentation with its fast, continuous, and high-precision (closed-loop resolution of 2 nm) position control. The piezo manipulator is controlled using a low-voltage piezo amplifier (E-503.00; PI GmbH & Co.) and a sensor-controller module (E-509.C3A; PI GmbH & Co.). An analog voltage output module (NI 9263, National Instruments (NI)) powers the piezo stage and moves the micro-indenter down until it contacts and indents the sample. An analog voltage input module (NI 9215, NI) monitors the positions as well as the forces. A technical drawing of the sensor holder is accessible on Github (see Data Availability) while the detailed description of the individual CFM components is provided in Vogler et al.[55]. Closed-loop control of the micro-indentation was implemented in LabVIEW™, which allowed us to set an indentation velocity (e.g., 4 μm/s) and a maximum force. Once a specimen is loaded into the setup and positioned appropriately, a single indentation takes ~10–15 s, which is mainly defined through the indentation velocity and the approach distance, i.e., the initial distance between the sensor and the specimen prior to being in contact (e.g., 20 μm). The ratio between force and deflection provides the stiffness, which first has to be corrected for the slight deformation of the measuring system itself. This is done by indenting a hard surface (e.g., glass slide) to calculate the systems stiffness ($k_{system}$). The corrected stiffness of the sample is then given by

$k_{sample} = \left(\frac{1}{k_{measured}} - \frac{1}{k_{system}}\right)^{-1}$. The MATLAB code used for the evaluation as well as example data are accessible on Github (see Code availability). Once a specimen is loaded into the setup and positioned appropriately, the experiments are monitored with an inverted microscope (IX71, Olympus K.K.), a ×20 objective lens (LUCPlanFL N x20; Olympus K.K.) and a CCD camera (Orca-D2; Hamamatsu Photonics K.K., Hamamatsu). For fluorescence imaging, an LED illumination system (pE-2, CoolLED) with an excitation wavelength of 480 nm was used to visualize the Green Fluorescent Protein (GFP) in the *C. elegans* transgenic line.

As the force sensor is positioned on the opposite side of the specimen with respect to the lens and is, therefore, not visible during the indentation process, it is crucial to determine the position of the force probe prior to the mechanical characterization. Once positioned, its location remains fixed with respect to the image frame, ensuring perpendicular indentations. Nevertheless, additional evaluations, such as a thorough analysis of the recorded force-displacement curve to detect slippage, are necessary to ensure proper interpretation of the results.

The repeatability of our mechanical characterization method ($n = 5$, $m = 50$, average coefficient of variation CV = 4.8%) regarding the quantification of biological specimens has been investigated using onion epidermis cells to prevent re-orientation of the sample during sensor movement. Detailed statistical analysis of the results (see Supplementary Fig. 1b) can be found in the Supplementary Note 2.

**Fabrication of rotational manipulation device**. A step-by-step protocol describing the fabrication procedure of the acoustic device can be found at Protocol Exchange[56]. Standard single layer photolithography processes are used to fabricate 500-μm-long SU-8 (MicroChem) microchannels (Supplementary Fig. 5) with a height and width of 150 μm and 80 μm, respectively. The channels are separated by 100 μm. Both specimen are manipulated using the same chip design. The mask design used for photolithography is available on Github (see Data Availability). Subsequently, the processed wafers are coated with silane (1H, 1H, 2H, 2H-

Perfluorooctyltriethoxysilane, abcr GmbH). The pattern is transferred into polydimethylsiloxane (PDMS, Sylgard 184, Dow Corning Inc.) and the channels are cut at right angles to create open microcavities. Then, the PDMS cavities are bonded to a glass slide by oxygen plasma treatment. A piezoelectric transducer (KPEG-126, Kingstate) is glued to the glass slide using two-component epoxy. By placing droplets of liquid (water or calcium solution) in front of the microcavities, microbubbles are simultaneously trapped inside the cavities due to hydrophobic/hydrophilic interactions. The microbubbles are acoustically excited via the transducer using an arbitrary function generator. The device is implemented into a CFM setup to perform micro-indentations (Supplementary Fig. 1a). Lily pollen grains are trapped near the surface of a single microbubble (Fig. 1b), while multiple parallel bubbles are used for the rotation of a *C. elegans* nematode (Fig. 1c). To prevent possible damage to the fragile MEMS force sensor as well as radiation force-based attraction between the sensor probe and the specimen during acoustic excitation, it is crucial to position the force sensor at a distance of at least 20 μm from the sample. However, the exact position might vary depending on the applied input power and the resulting strength of the acoustic field.

Due to its open structure, and the chemical stability of PDMS, the device can be re-used after cleaning with common solvents such as ethanol or isopropanol.

While our design allowed for different shapes of micro-vortices, we opted to use out-of-plane vortices to rotate the specimen along the *y*-axis (Fig. 1b, c), because it allows us to expose parts of the specimen that were previously obstructed as well as due to its strong presence throughout all investigated excitation frequencies.

**Plant model**. Anthers of *Lilium longiflorum* (lily) containing the pollen grains were collected and stored in Eppendorf tubes at −80 °C. An hour before the experiment, the Eppendorf tubes were removed from the freezer, opened and placed in an incubation chamber to rehydrate the pollen grains. A small amount of pollen was transferred into a new Eppendorf tube and the pollen grains were submerged and mixed with either deionized water or a 5 mM CaCl$_2$ solution. For the rotational indentation experiments, the submerged lily pollen grains were positioned dropwise near the PDMS structure.

For experiments requiring non-hydrated lily pollen grains, the Eppendorf tubes containing the frozen plant cells were removed from the freezer and kept open at room temperature for at least 30 min. The pollen grains have time to thaw; however, they do not hydrate due to the low surrounding humidity. The lily pollen grains were then spread on a glass slide covered with a thin layer of silane (#63411-01, Science Services) to reduce slippage during indentation.

Hydrated lily pollen grains have a major axis of 128.5 ± 9.9 μm and a minor axis of 98.3 ± 5.8 μm[54]. Since non-hydrated pollen grains are folded, they exhibit a comparatively smaller minor axis.

For the quantification of hydrated and non-hydrated pollen grains a maximum force of 10 μN was applied.

**Numerical modeling**. Indentations of pollen grains have been reproduced using FEM-based simulations built upon MorphoMechanX (http://www.morphomechanx.org). The initially unpressurized pollen grain has a major and minor axis of 128 μm and 97 μm, respectively, which, if pressurized, resembles the dimensions of an average lily pollen grain (128.5 μm and 98.3 μm). Based on average data from the literature[57], the intine thickness was assigned to be 1.5 μm and the exine 0.5 μm. After inflation, the bottom half of the template was fixed by Dirichlet boundary conditions and indentations were performed as described in Mosca et al.[30]. The pollen grain is represented by means of multi-layered solid (wedge) elements to reproduce the differential contribution of intine and exine to the indentation curves. Given the non-negligible thickness of the intine, solid elements were considered optimal to reliably account for in-thickness compression of the cell wall. Further information is provided in the Supplementary Note 6 and refs. [58,59].

The simulated pollen grain (see Supplementary Fig. 4a, b) was inflated to different pressure values (0.1, 0.2, and 0.3 MPa) and different Young's moduli were assigned to the intine ($E_i$ = 5–100 MPa) and exine ($E_e$ = 10–200 MPa). Indentations were performed at the center of the colpus for intine measurements as well as opposite of the colpus for exine characterizations (see Supplementary Fig. 4c). The apparent stiffness values for intine and exine measurements were computed at indentation depths of 1.6 μm and 0.6 μm, respectively, and are presented in Supplementary Table 1 for various combinations of material properties.

To investigate the origin of the sublinear indentation curves, a spherical shell model with a diameter of 100 μm, i.e., comparable to the dimensions of a lily pollen grain, has been implemented. The shell thickness was varied across 0.5 and 1.5 μm for the simulation with wedges and 0.2 μm for membrane simulations. The results of each indentation are presented in Supplementary Fig. 4g, including a linear fit (fitting range = indentation depth of 0.1–0.5 μm) to demonstrate the corresponding superlinear and sublinear behavior of the force-displacement curves.

**Animal model**. Day-2 adult *C. elegans* were cultured and maintained at 20 °C on nematode growth media (NGM) plates using standard protocols[60], while being fed with standard E. coli OP50 lawns. The *C. elegans* strain TJ375 (P*hsp-16.2*::GFP)[61] from the *Caenorhabditis* Genetics Center of the University of Minnesota was used, because of the high expression of GFP in the animal intestine after 1 h at 37 °C

heat-shock. Twelve hours prior to each experiment, a population of TJ375 was heat-shocked, and immediately before manipulation they were washed out of the NGM plates with M9 buffer. In order to prevent motion during indentation, we used 1 mM of the drug levamisole (L0380000, Sigma-Aldrich)[62], which is an acetylcholine agonist, that induces paralysis and muscle contraction in *C. elegans*. Nematodes were exposed 5–10 min to the drug before being mounted onto the manipulation stage.

During the experimental procedure, the force sensor was moved along the longitudinal axis of the *C. elegans* nematode. The stiffness measurements presented in Fig. 3f do not follow a straight line as the position of the indentation has been corrected to account for rotational movements of the specimen during mechanical characterization.

**Statistics and reproducibility**. All data on pollen grains are based on repeated experiments on independent samples and the components of the individual box-plots are expressed in the legends. Data characterizing nematodes were collected from a single specimen through independent measurements. For datasets with $m \leq 10$ measurements, each data point is denoted by a dot. Where appropriate, data were tested on normality, and F-test as well as double-sided t-tests were performed to compare datasets. Microsoft Excel (Microsoft Corporation), GraphPad Prism (GraphPad software), and IBM SPSS Statistics (IBM) were used for statistical analysis and data handling. Detailed results of the statistical evaluations are presented in the Supplementary Notes 2, 3, 4, and 5. *p*-values of <0.05 were considered as statistically significant.

All presented micrographs were repeatable for different samples in independent experiments.

**Ethical approval**. We affirm that we have complied with all relevant ethical regulations for animal testing and research. Given that our experiments focused exclusively on *L. longiflorum* pollen grains, *C. elegans* nematodes, and onion epidermal cells, no ethical approval was required for any of the presented work.

**Reporting summary**. Further information on research design is available in the Nature Research Reporting Summary linked to this article.

## Data availability

The authors declare that data supporting the findings of this study are available within the paper and its supplementary information. Source data are provided with this paper and are, additionally, available on Github together with technical drawings and information on the design of the acoustic device:[63] https://github.com/laeublin/3D-Indentation. Further data can be accessed through the authors.

## Code availability

The MATLAB code required for the evaluation of the micro-indentations, appropriate example data, as well as information on the expected output is available on Github:[63] https://github.com/laeublin/3D-Indentation. The model used for indentation simulation of pollen grains is available on Github:[64] https://github.com/GabriellaMosca/PollenGrain_indentation. To run such a model, MorphoMechanX is required and can be obtained upon request (www.morphomechanx.org). For ease of visualization, vLab is proposed (http://algorithmicbotany.org/vlab/).

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

## Acknowledgements

The authors thank Daniel Bollier (University of Zurich) for the fabrication as well as the technical drawing of the custom sensor holder. This work is supported by the ETH Zurich, the University of Zurich, and, in part, by a grant from SystemsX.ch, the Swiss Initiative in Systems Biology (Research and Technology Development Project MecanX) to U.G and B.J.N., an interdisciplinary grant from the Swiss National Science

Foundation (Grant Number CR22I2_166110) to U.G. and B.J.N., and the ETH Zurich Career Seed Grant-14 17-2 to D. A.

## Author contributions

N.F.L., N.S., and D.A. conceived the initial idea. B.J.N., N.S, U.G., and D.A. supervised the project. B.J.N., U.G., and D.A. raised funding. N.F.L. designed and fabricated the manipulation device. J.T.B. wrote the control and readout software for indentations. H. V., and N.V.-Q. cultivated and prepared the samples. N.F.L., J.T.B., J.M., and N.V.-Q. performed and evaluated the experiments. G.M. developed the numerical model and designed and performed the indentation simulations. D.A., H.V., and A.dM. provided know-how and suggestions. N.F.L., J.T.B., and N.V.-Q. wrote the manuscript with help from D.A., N.S., H.V., U.G., and G.M. All authors reviewed and commented on the manuscript.

## Competing interests

The authors declare no competing interests.
