## [Peer Review File · Nature Communications]

Reviewers' Comments:

Reviewer #1:

Remarks to the Author:

I very much enjoyed reading this article by Läubli et al. They have developed a novel microfluidics-acoustic system for the stabilization and manipulation of biological samples that is compatible with microindentation techniques. It is a very interesting idea and it appears to work very well for the types of experiments that the authors are targeting. I think there is an under appreciation for how important this type of tool is in mechanical measurements. When I am asked if I can make a measurement on tissue X, my first reply is always "can you hold it down well enough to make a decent measurement?". Here the authors appear to have not only succeeded in that, but have also devised a way to rotate the samples very precisely as well.

As I work mostly with plants, I will comment mainly on the plant measurements. Although the method is limited to small samples, for example it is not going to work with a leaf, meristem or sepal, it still has quite some range from microns to a millimeter or so. It would definitely work with plant culture cells, or small dissected tissue.

The main criticism I have is not related to the technology itself, but rather to the interpretation of the results on the lily pollen grains. I think the authors need to put their observations in context with previous measurements on plant tissue. The pollen grain diameter ($\sim 30\mu\text{m}$) is similar in diameter to the onion cells measured in Routier-Kierzkowska et al. 2012 (Plant Phys 158), and the stiffness measured for turgid onion cells was almost the same at $\sim 16\text{N/m}$. Similar values are recorded for turgid BY2 cells (Weber et al. 2015, JXB 66), which are also a similar radius. Both papers show that most of the measured "stiffness" for indentation at this scale is from the turgor pressure and geometry, rather than the cell wall material properties. Note that both these papers use the same CFM technology (FemtoTools force sensors) developed in the author's (Nelson) lab.

The authors suggest that they are measuring the stiffness of the cell wall in the exine and intine. If the grains have a normal turgor pressure similar to that of BY2 and onion cells, then 14N/m as measured for the exine is reasonable, even if the walls were very soft. If the exine wall really were very stiff, then the measurement should be higher, because it can't really go any lower than the stiffness induced by the turgor.

In addition to turgor pressure, Weber et al. show that the geometry of the sample greatly affects the measurement. A larger cell will feel stiffer even if the turgor pressure and wall stiffness is the same. Beauchamy et al. 2015 (Biophysical Journal 108) present some nice work showing how the measured stiffness changes based on the curvature of the sample. It may be especially relevant here, as the intine and exine look to have substantially different curvature. The intine is much more curved, so one would expect the apparent stiffness to be lower, even if the turgor pressure and wall material properties are the same.

One final point is that for small samples the indentation will cause a deformation on the opposite side of the structure. This needs to be taken into account if one wants the absolute indentation stiffness, whereas for comparative measurements, such as with or without calcium, this is likely not much of an issue.

The measurable stiffening of the pollen grain in response to calcium is nice, although one needs to be careful to control for shape or turgor induced changes there.

For all measurement values reported in the text, it would be nice to have the number of repeats and the standard deviation or error.

The code, data, any drawings and/or meshes needed to fabricate the microfluidics, sensor holders, etc. should be put online somewhere.

Page 2:

- Routier-Kierzkowska et al. 2012 (Plant Phys 158) really should be referenced for the CFM, after all they coined the term.

Page 5:

- If you look from below, how can you tell if you are indenting on a flat part at the top? I suppose if the surface is sloped, you would get a lower stiffness measurement (see Routier-Kierzkowska et al. for an analysis of this effect). This is a bit of a concern here, as the pollen grain has substantial curvature.

- Do you know how much the grain will rotate with each pulse? Is it consistent? Or is there significant variability there?

- Fig 2f shows an indentation distance of ~1.5 microns, so it is likely you mostly feel turgor pressure with these experiments, but you claim it is cell wall properties (bending stiffness?). Does this really fit with previous results?

Page 8

- the authors write "This indicates that the hydration phase causes extreme changes in the cell wall matrix, as the applied forces are small enough to prevent compression of the cell and only provide local material properties" Are you sure this is true? I think I believe that there are changes in the wall with hydration, but I am not sure they are mostly what you are measuring with this experiment. Imagine a balloon filled with dry mud. It would get much softer if I added some water inside, even though the wall material is the same. It might be better to just say that the apparent stiffness decreases upon hydration.

Reviewer #2:

Remarks to the Author:

The authors describe a tool to measure the mechanical properties of biological samples at various hierarchical biological scales (cell, tissue, and organismal). Using a combination of acoustic manipulation, and micro-indentation the tool provides manipulation capabilities that allow access to multiple regions of a biological sample. The tool is more general, and not custom designed for any one type of specimen. The authors leverage the length scales (low Reynolds number prevents drift) to obtain precise control while positioning the specimen during rotation.

The authors demonstrate the utility of this tool by measuring the mechanical properties of two significantly different specimens. They measure the stiffness of pollen grains that are dehydrated, and rehydrated, and in the presence of CaCl_2 (a crosslinking trigger) and find significant differences in stiffness both within each sample (between the intine and the exine) and between samples under varying conditions. Secondly, they measure the stiffness of an adult *C. elegans* worm at different locations on the body using the indentation technique concluding that variations in body stiffness along the body arise from differences in internal body structures. Overall, the methods section is well written, and easy to follow. The abstract, however, is a little vague. Suggestions to improve the abstract and some major points of the paper are given below. To my knowledge this is the first demonstration of 3D acoustic manipulation being combined with micro-indentation for mechanical measurements of biological samples. However, I would like to note that neither 3D acoustic manipulation (the authors themselves have published the same protocol - D. Ahmed et al, 2016) nor micro-indentation (for example CFM has been used for pollen tubes, again by B. Nelson's group and others) are novel by themselves and have been leveraged individually in many previous studies as the authors themselves have stated.

Given the potential utility of this tool, I would have thought the authors would leverage it to develop a more cohesive set of results on either one of the systems (i.e pollen grains or *C. elegans* worms) and push further than being a simple demonstration of combining two well known techniques. On this front, I would rate the novelty relatively low for Nature communications. As indicated below, the data they present already includes some interesting trends that could be investigated further.

However, the techniques presented here do offer several advantages for stiffness measurements of biological samples. The fact that the technique is not specimen dependent and that it appears relatively easy to implement allows it be more general and relevant for a very wide audience.

Comments on the Statistics

- The addition of CaCl_2 leads to “an intine stiffness which is on average 0.7 times the stiffness value of the exine.” What is the error on that value? Is it really significant as compared to the 0.56 in the previous ‘dry’ experiments? When looking deeper, one finds in the SI that the precise value is 0.66. I have two comments regarding the statistics here:
 - o Both intine and exine data look bimodal – there might be something of interest there – I would advise the authors to look more deeply at their data.
 - o While the data may be statistically significant I am not sure these small differences are really significant. At the very least, using a non-parametric test might be better since we cannot assume a normal distribution (unless authors have verified that the distribution of means is normally distributed).
 - o I hope the authors can rewrite this section more accurately and include more details.
- For the measurement of the dehydrated pollen grain in SF3 – taking the mean of the measurements is not a good representation of the distribution (given that it looks nearly bimodal).
- In Fig 3F, could the authors state how many worms were used to determine that the body wall muscle leads to higher stiffness measurements?
- Fig SF3 claims $n=300$ for the pollen grain in CaCl_2 in panel b, is this supposed to be $n=30$ as stated in the text and caption?

Other Suggestions

- C. Essmann et al. Nat Comm Feb 2020 should be cited
- In the abstract, it would be useful if the authors specifically mention the results obtained from the experiments (for example that measuring stiffness at two different positions on the worm revealed differences).
- The authors make the claim that they “demonstrate the utility of [their] platform for measurements at the organism level by detecting the influence of internal organs on the mechanical properties of surrounding tissue in different regions of individual nematodes.” For the sake of precision, I do not think their results support this claim. Rather they have shown differences in mechanical properties at two individual locations on the worm body. Whether this difference really arises from the influence of internal organs or from local variations on the cuticle itself is not clear.
- The authors claim methods such as glue for immobilizing worms is invasive, but there is strong evidence that chemically paralyzing a worm is also invasive. As the authors themselves point out it has an effect on the body wall muscle and therefore may alter the mechanical properties from the natural state. It is inaccurate therefore to imply that paralyzing the worm is non-invasive.
- For readability:
 - i. It would be useful for authors elaborate on how the acoustically actuated microbubbles are produced in the main text, though some of it is included in the SI.
 - ii. It is not clear how the specimen is delivered to the manipulation device – in a close microfluidic device, flow is used. In this open system, is it simply placed close to the bubbles? Please specify
 - iii. Is the piezo positioning stage used in combination with microscopy to position the force sensor? Please clarify in text.
 - iv. Please specify how long each measurement requires (for methods section)
 - v. Pg 9. A clarification is required in the statement: “the less soft regions are directly above the eggs/embryos inside the gonad (where the dorsal muscles are not present), while stiffer measurements were recorded in the region of the body wall muscles”. Do they mean soft or less soft? I am assuming less soft means stiff! This might just be a typo.
 - vi. Since this is a methods paper, please include the schematic/photograph of the setup in the main text Fig1 rather than in the SI

Reviewer #4:

Remarks to the Author:

3D Mechanical Characterization of Single Cells and Small Organisms using a Combination of Acoustic Manipulation and Force Microscopy

In this article the authors report on the combination of an acoustically driven manipulation device to trap and rotate biological samples and a micro-indenter. The device consists of a piezoelectric transducer mounted on a glass slide together with a PDMS structure. The main principle is given by microbubbles caught in the cavities of this structure. These microbubbles are excited acoustically leading to the well known acoustic streaming. As inertia only plays a minor role, this allows to precisely rotating the trapped object as desired.

The structure of the article is clear and all figures are very carefully and nicely prepared. The methods seem solid to the widest extent (minor suggestions see below).

However, I find the article is not suited for publication in nature communications due to the following reasons:

- The study is a basic combination of earlier work of the group (force microscopy+acoustic rotation).

Force Microscopy

Vogler et al. / The Plant Journal(2013)73,617–627

Shamsudhin et al. / PLOS ONE | DOI:10.1371/journal.pone.0168138 December 15, 2016

Burri et al. / IEEE ROBOTICS AND AUTOMATION LETTERS, VOL. 4, NO. 2, APRIL 2019

Acoustically induced rotation:

Nino Läubli et al. / Procedia CIRP 65 (2017) 93 – 98,

Nino F. Läubli et al. / Small Methods 2019, 3, 1800527

Thus the advance for me seems too incremental to be suited for Nature communications.

- Additionally, it somehow irritates me, that the original work of the group presenting the rotation method (Nino F. Läubli et al. / Small Methods 2019, 3, 1800527) is very close to work published already by one of the authors in 2016 in NC (Ahmed et al. NATURE COMMUNICATIONS | 7:11085 | DOI: 10.1038/ncomms11085).

- Many claims (e.g. "Acoustofluidic devices generally consist of closed microchannels") are probably not really justified, which might be caused by a lack of knowledge of existing literature from the early 2000s. This lack also affects the reference list, where some fundamental pioneering papers are missing.

Nevertheless, the manuscript seems publishable in another journal claiming less ground breaking advance with only minor modifications.

For this purpose, I would recommend to apply a model (Hertz?) for the indentation measurements to present quantitative values of the Young's modulus.

Moreover, I would recommend to supply some additional statistical test (t-test or other suitable) to additionally support the biological findings.

In summary, I do not recommend publication of this work in Nature Communications for the reasons given above. It does not meet the criteria.

Detailed Response to the Reviewers' Comments

First, we would like to thank the reviewers for their insightful and constructive criticisms. We have tried to address all the points they made in the revised version of our manuscript. Below you can find a point-by-point response to the reviewers' comments. For ease of reading, the reviewers' comments are shown in blue and our responses in *italics*.

Reviewer #1 (Remarks to the Author):

I very much enjoyed reading this article by Läubli et al. They have developed a novel microfluidics-acoustic system for the stabilization and manipulation of biological samples that is compatible with microindentation techniques. It is a very interesting idea and it appears to work very well for the types of experiments that the authors are targeting. I think there is an under appreciation for how important this type of tool is in mechanical measurements. When I am asked if I can make a measurement on tissue X, my first reply is always "can you hold it down well enough to make a decent measurement?". Here the authors appear to have not only succeeded in that, but have also devised a way to rotate the samples very precisely as well.

As I work mostly with plants, I will comment mainly on the plant measurements. Although the method is limited to small samples, for example it is not going to work with a leaf, meristem or sepal, it still has quite some range from microns to a millimeter or so. It would definitely work with plant culture cells, or small dissected tissue.

Response:

We thank the reviewer for his/her supportive assessment and for highlighting the importance of the new technique presented.

The main criticism I have is not related to the technology itself, but rather to the interpretation of the results on the lily pollen grains. I think the authors need to put their observations in context with previous measurements on plant tissue. The pollen grain diameter (~30µm) is similar in diameter to the onions cells measured in Routier-Kierzkowska et al. 2012 (Plant Phys 158), and the stiffness measured for turgid onion cells was almost the same at ~16N/m. Similar values are recorded for turgid BY2 cells (Weber et al. 2015, JXB 66), which are also a similar radius. Both papers show that most of the measured "stiffness" for indentation at this scale is from the turgor pressure and geometry, rather than the cell wall material properties. Note that both these papers use the same CFM technology (FemtoTools force sensors) developed in the author's (Nelson) lab.

Response:

We thank the reviewer for sharing his/her concern regarding the mechanical characterization and the interpretation of the results. Based on the reviewer's valuable input concerning apparent stiffness and its relation to material stiffness, we have now revised the manuscript significantly to ensure proper evaluation and interpretation of the results. Additionally, a numerical simulation has been performed to shed light on the complex relationship between turgor pressure, specimen geometry, and apparent stiffness. However, it is important to highlight that, in contrast to pollen tubes (as well as onion cells or BY2 cells), the pollen grain diameters are approximately 100 µm (minor axis) and thus the large difference in curvature will have a distinct influence on the apparent stiffness than in the stated examples. The precise characterization of lily pollen grain dimensions, based on previous publications, is also presented in the Materials and Methods section. Nevertheless, the reviewer's concerns have been analyzed in detail and adjustments to the manuscript are highlighted in the following responses. To prevent any confusion, the approximate dimensions of pollen grains have been added to the Results section of the manuscript (line 160), i.e. "We used the 3D acoustic manipulation device combined with CFM to quantify the apparent stiffness of a hydrated pollen grain (major axis of 128.5±9.9 µm; minor axis of 98.3±5.8 µm) at different surface regions."

The authors suggest that they are measuring the stiffness of the cell wall in the exine and intine. If the grains have a normal turgor pressure similar to that of BY2 and onion cells, then 14N/m as measured for the exine is reasonable, even if the walls were very soft. If the exine wall really were very stiff, then the measurement should be higher, because it can't really go any lower than the stiffness induced by the turgor.

Response:

We thank the reviewer for raising this point. First of all, we would like to highlight that the different contributions of turgor pressure as well as material stiffness may be substantially changed due to the larger dimensions of pollen grains compared to the cells mentioned. Nevertheless, the presented relation has been investigated through numerical simulations of microindentations on pollen grains for various combinations of turgor pressures with intine and exine material properties (see Supplementary Table ST1). These investigations reveal that the observed apparent stiffness values could be reproduced for a turgor pressure of 0.2 MPa with Young's moduli $E_i = 10$ MPa for the intine and $E_e = 100$ MPa for the exine. This suggests that for stiff materials such as the exine, a higher apparent stiffness is not necessarily expected.

Based on the reviewer's input, this conclusion has now been added to the manuscript (line 235), i.e. "When arriving at a ratio of 10, i.e., $E_e = 100$ MPa and $E_i = 10$ MPa, the exine's apparent stiffness (see Supplementary Table ST1) reaches a value of 12.7 N/m (experimental $k_e = 14.3$ N/m; relative difference 11.8%), indicating that an appropriate material stiffness ratio E_e / E_i is around 10 for the experimentally determined apparent stiffness ratio. ... While changes in exine material stiffness display only a small effect on the simulated properties of the intine, the resulting apparent stiffness value, i.e., $k_i = 7.9$ N/m, moved closer to our experimental measurements."

In addition to turgor pressure, Weber et al. show that the geometry of the sample greatly affects the measurement. A larger cell will feel stiffer even if the turgor pressure and wall stiffness is the same. Beauzamy et al. 2015 (Biophysical Journal 108) present some nice work showing how the measured stiffness changes based on the curvature of the sample. It may be especially relevant here, as the intine and exine look to have substantially different curvature. The intine is much more curved, so one would expect the apparent stiffness to be lower, even if the turgor pressure and wall material properties are the same.

Response:

We thank the reviewer for sharing his/her concern regarding the influence of the curvature on the detected apparent stiffness values for the intine as well as the exine. While it is correct that curvature can have a substantial effect on indentation-based characterizations, it is important to highlight that the used indenter (tip diameter $< 2 \mu\text{m}$) is significantly smaller than the specimen (minor diameter $\approx 100 \mu\text{m}$) and, therefore, the indentation area (if near the center axis of the specimen) can be assumed to be locally flat and orthogonal to the measurement axis. Nevertheless, any indentation process, especially for large forces, is likely to result in deformation of the surrounding tissue and, thus, will be affected by the curvature of the specimen. Therefore, we investigated the curvature of the intine and exine for the proposed combination of material stiffness and turgor pressure for which the simulated apparent stiffness values were well aligned with our experimental data. Supplementary Fig. SF4b shows the simulated pollen grain prior to being pressurized as well as the outline of the pressurized pollen grain, which corresponds to the hydrated pollen grain (see below). While a small change in curvature can be observed between the exine and the intine, based on the lower material stiffness of the intine, the resulting small difference can be assumed to be of minor importance for the observed variation in apparent stiffness, especially in comparison to the underlying material stiffness ratio E_e/E_i of 10.

Supplementary Fig. SF4b:

That said, the importance of additional effects, such as curvature on mechanical characterizations has been highlighted in the manuscript (line 199), i.e., “The apparent stiffness obtained from experimental indentation is, as shown in previous work, strongly affected by various parameters, such as local curvature, cell size, material properties, or turgor pressure.”

One final point is that for small samples the indentation will cause a deformation on the opposite side of the structure. This needs to be taken into account if one wants the absolute apparent stiffness, whereas for comparative measurements, such as with or without calcium, this is likely not much of an issue. The measurable stiffening of the pollen grain in response to calcium is nice, although one needs to be careful to control for shape or turgor induced changes there.

Response:

We thank the reviewer for his/her input. We agree that some slight deformation, especially if the exine is indented opposite to the intine, cannot be fully prevented in indentation-based characterizations and would likely influence the corresponding measurement to a certain extent. This can even be the case for our characterization method, where only small forces are applied to a local point of contact and global deformations are not observed. Therefore, we decided to simulate the exine indentation with the aim to reproduce the corresponding factor in our investigation and derivation of the pollen grain’s material stiffness. As also mentioned by the reviewer, it is important to highlight that for the evaluation of the environmental influence, a stiffness ratio for each individual pollen has been calculated and that the additional deformation on the opposite side probably does not matter for the resulting comparison and statistical evaluation. Furthermore, according to the reviewer’s suggestion, the terminology has been adjusted significantly throughout the manuscript to prevent possible confusion between “apparent stiffness” and “material stiffness”.

Additionally, we added the following sentence to the discussion to provide further possible explanations for the observed change in stiffness ratio (line 397), i.e., “... observed significant variations induced by the liquid environment, although their exact source, e.g., changes in material stiffness, turgor pressure, or pollen grain geometry, remains to be determined.”

For all measurement values reported in the text, it would be nice to have the number of repeats and the standard deviation or error.

Response:

The measurements on the pollen have been performed using single point indentations, i.e., a single measurement per position. This is due to the difficulty of stabilizing the sample,

since movement of the sensor in the nearby liquid may slightly alter the pollen's orientation. Given the high variation of surface features on the pollen grain's exine with the nexine at the lumina being closely surrounded by the sexine, a repeated experiment could unintentionally lead to measurement of a different material, which would significantly reduce accuracy and not reflect the measurement error but rather variations of the specimen's surface structure.

However, given the importance of the reviewer's request, an additional experimental series ($m = 50$) has been performed on onion epidermal cells to quantify the repeatability of our characterization approach for biological specimens. It should be noted that onion epidermal cells were chosen due to their homogeneous cell wall and to prevent the possible influence of re-orientations (based on sensor movement and subsequently produced microstreaming) on the mechanical quantifications.

Following the reviewer's input, the following sections have been added to the manuscript (line 136 and line 478), i.e., "The technique shows a high repeatability for the mechanical characterization of biological specimens with an average coefficient of variation of 4.8% ($n = 5$, $m = 50$, see Supplementary Fig. SF1b and Supplementary Information SI2). Please note that the derived reproducibility can also be affected by local changes in the biological specimen induced through repeated indentations."

"The repeatability of our mechanical characterization method ($n = 5$, $m = 50$, average coefficient of variation $CV = 4.8\%$) regarding the quantification of biological specimens has been investigated using onion epidermis cells to prevent re-orientation of the sample during sensor movement. Detailed statistical analysis of the results (see Supplementary Fig. SF1b) can be found in the Supplementary Information SI2."

And in Supplementary Material (line 19 and line 169) the following was added:

"Measurements on onion epidermal cells used to quantify the repeatability of our mechanical characterizations of biological specimens. The graph shows the apparent stiffness values (with Tukey-style whiskers) for 5 different onion cells with 10 measurements each. The resulting average coefficient of variation CV is 4.8%. A detailed statistical evaluation is provided in Supplementary Information SI2."

"To quantify the repeatability of our indentation-based mechanical characterizations, 50 measurements were performed on 5 onion epidermal cells, i.e., 10 repeated indentations per cell (see Supplementary Fig. SF1b). Onion epidermal cells were chosen as they have a homogeneous cell wall and prevent possible measurement artifacts through realignment or slippage. Repeatability for each sample was derived through the coefficient of variation (CV) according to B. Carstensen. Chapter 9: Repeatability, Reproducibility and Coefficient of Variation. In *Comparing Clinical Measurement Methods: A Practical Guide* (2010). The individual CV s for the cells 1 – 5 are 6%, 3%, 5.1%, 3.5%, and 6.6%, respectively, which leads to an average CV of 4.8%.

It is crucial to highlight that the reproducibility of mechanical characterizations is also affected by changes in the biological specimen, such as local cell wall weakening, induced through the repeated indentations or changes in turgor pressure due to the hydration state of the sample."

The code, data, any drawings and/or meshes needed to fabricate the microfluidics, sensor holders, etc. should be put online somewhere.

Response:

We thank the reviewer for this input. This is an important aspect to simplify the adaption of our setup by interested readers. Therefore, the technical drawing of the sensor holder as well as the mask design used for the fabrication of the acoustic device have been made

available through Github. Furthermore, code and example data used for the evaluation of the indentation measurements is provided in the repository. Extensive documentation relating to the setup and its additional components is also provided in Vogler et al. Simultaneous measurement of turgor pressure and cell wall elasticity in growing pollen tubes, in *Plant Cell Biology* (2020) and the corresponding information has been added to the manuscript to make future readers aware of the availability of this information. Additionally, all data required for the in-depth investigation of plant cell indentation through numerical simulations has been added to a separate Github repository.

Per the reviewer's suggestion, in addition to setting up the Github repository, the following parts have been added to the manuscript (line 450, line 461, line 488, line 590, and line 596), i.e.

"A technical drawing of the sensor holder is accessible on Github (see Data Availability) while the detailed description of the individual CFM components is provided in Vogler et al." "MATLAB code used for the evaluation as well as example data are accessible on Github (see Code Availability)."

"The mask design used for photolithography is available on Github (see Data Availability)."

"Source data are provided with this paper and are, additionally, available on Github together with technical drawings and information on the design of the acoustic device:

<https://github.com/laeublin/3D-Indentation>"

"The MATLAB code required for the evaluation of the microindentations, appropriate example data, as well as information on the expected output is available on Github:

<https://github.com/laeublin/3D-Indentation>

The model used for indentation simulation of pollen grains is available on Github:

https://github.com/GabriellaMosca/PollenGrain_indentation

To run such a model, MorphoMechanX is required and can be obtained upon request

(www.morphomechanx.org). For ease of visualization, vLab is proposed

(<http://algorithmicbotany.org/vlab/>)."

Page 2:

- Routier-Kierzkowska et al. 2012 (*Plant Phys* 158) really should be referenced for the CFM, after all they coined the term.

Response:

We fully agree that this and the work by Weber et al. are relevant publications concerning the mechanical quantification of plant specimens through micro-indentation and they have, together with other studies on the indentation of shell-type plant specimens, been added to the references, i.e.

Routier-Kierzkowska et al., *Plant Physiol.* (2012) **158**

Weber et al., *J. Exp. Bot.* (2015) **14**

Vella et al., *J. R. Soc. Interface* (2012) **9**

Vella et al., *Phys. Rev. Lett.* (2012) **109**

Page 5: - If you look from below, how can you tell if you are indenting on a flat part at the top? I suppose if the surface is sloped, you would get a lower stiffness measurement (see Routier-Kierzkowska et al. for an analysis of this effect). This is a bit of a concern here, as the pollen grain has substantial curvature.

Response:

We thank the reviewer for his/her valuable input as it is correct that curvature can significantly affect the detected apparent stiffness due the sensor's limitation of measuring forces in a single direction. While the sensor probe is opposite to the objective lens with respect to the pollen grain and, thus, invisible, its position is known due to the previous indentation of the glass slide (required to quantify the system's internal stiffness). As the

sensor's position remains fixed within the microscope's field of view, this allows us to move near the center axis of the specimen where, given a tip diameter of less than 2 μm and a specimen size of more than 100 μm , a nearly flat tissue can be assumed. Furthermore, if the specimen was not indented perpendicularly but at a certain angle, slippage would occur, which would be visible in the recorded force-displacement curve. It is, therefore, important to highlight that all data were assessed for such artifacts and only measurements that showed no slippage were evaluated and included in this manuscript.

To reflect this issue with indentation-based characterizations and our corresponding measures, the following sentence has been added to the manuscript (line 470), i.e. "As the force sensor is positioned on the opposite side of the specimen with respect to the lens and is, therefore, not visible during the indentation process, it is crucial to determine the position of the force probe prior to the mechanical characterization. Once positioned, its location remains fixed with respect to the image frame, ensuring perpendicular indentations. Nevertheless, additional evaluations, such as a thorough analysis of the recorded force-displacement curve to detect slippage, are necessary to ensure proper interpretation of the results."

- Do you know how much the grain will rotate with each pulse? Is it consistent? Or is there significant variability there?

Response:

While our acoustic-based manipulation allows for high control over the specimen's rotational velocity (as can be seen in Supplementary Figure SF2c), its motion can differ depending on various factors, including the specimen's size and weight distribution as well as the dimensions of the microbubble and its location with respect to the piezoelectric transducer. To prevent the necessity of analyzing each rotational pattern, which may also vary over time due to the rectified diffusion, the specimen's rotation was terminated manually. Despite this non-automated approach, given the very slow yet stable rotations possible at low excitation voltages, the position of indentation, and by that the orientation of the sample, can be chosen very precisely to allow for complete coverage of the cell during the subsequent mechanical characterization.

Per the reviewer's comment and to improve accessibility, the following sentence has been added to the manuscript (line 122): "As the rotational velocity can vary slightly depending on the specimen as well as the microbubble, the rotation of the sample was stopped manually."

- Fig 2f shows an indentation distance of ~1.5 microns, so it is likely you mostly feel turgor pressure with these experiments, but you claim it is cell wall properties (bending stiffness?). Does this really fit with previous results?

Response:

We thank the reviewer for sharing his/her concern regarding the indentation depth and the resulting quantified properties of the plant specimens. We would like to note that the indentation distance is in the same range as the cell wall thickness. Nevertheless, it is, of course, absolutely correct that during our mechanical characterizations, only the apparent stiffness is detected, which includes properties of the cell wall as well as the internal turgor pressure.

The notation has been adjusted throughout the manuscript to prevent possible confusions between apparent stiffness and material stiffness. For example, the following sentence has been added to the manuscript to introduce the concept of apparent stiffness (line 134): "The apparent stiffness contains local information on various specimen components and thus is, in plant cells, influenced by the mechanical properties of the cell wall, the internal turgor pressure, as well as geometric factors."

Furthermore, the effect of the individual components has been disentangled by using an in-depth numerical simulation that allows further discussion of our results as well as indentation-based measurements in general. It is worth noting that, despite the strong influence of the turgor pressure, the model revealed that a change in cell wall material, as is the case for the intine compared to the exine, leads to a significant change in the detected apparent stiffness for the indentation depths applied in our experimental investigations.

The model is introduced in a new section called Numerical Simulation of Plant Cell Indentation (line 198) as well as discussed in Materials and Methods (line 537), Discussion (line 387), and Supplementary Material through the means of Figures (Supplementary Fig. SF 4), Tables (Supplementary Table ST1), and Supplementary Information (Supplementary Information SI6, line 208).

Page 8

- the authors write "This indicates that the hydration phase causes extreme changes in the cell wall matrix, as the applied forces are small enough to prevent compression of the cell and only provide local material properties" Are you sure this is true? I think I believe that there are changes in the wall with hydration, but I am not sure they are mostly what you are measuring with this experiment. Imagine a balloon filled with dry mud. It would get much softer if I added some water inside, even though the wall material is the same. It might be better to just say that the apparent stiffness decreases upon hydration.

Response:

We thank the reviewer for his/her input regarding the measurements on non-hydrated pollen grains and the corresponding interpretation of the results. While we agree that there are certainly changes in the cell wall matrix induced by the hydration of the plant specimen, it is correct that our interpretation only focused on a single explanation for the detected difference in the cell wall matrix.

Therefore, we changed this claim and extended our interpretation as well as the discussion of this result to highlight additional theories. We adjusted the corresponding section as follows (line 288 and line 399): "This suggests that the hydration phase causes significant changes in the cell wall matrix, e.g., through swelling. However, additional factors, such as solid intracellular features or more stable configurations of the cell wall cannot be excluded." "Lastly, by characterizing the exine structure of non-hydrated pollen grains, we revealed that their apparent stiffness values decrease upon hydration while the complex relationship between intracellular features, exine material stiffness, as well as cell wall configuration and curvature remains unclear."

Reviewer #2 (Remarks to the Author):

The authors describe a tool to measure the mechanical properties of biological samples at various hierarchical biological scales (cell, tissue, and organismal). Using a combination of acoustic manipulation, and micro-indentation the tool provides manipulation capabilities that allow access to multiple regions of a biological sample. The tool is more general, and not custom designed for any one type of specimen. The authors leverage the length scales (low Reynolds number prevents drift) to obtain precise control while positioning the specimen during rotation. The authors demonstrate the utility of this tool by measuring the mechanical properties of two significantly different specimens. They measure the stiffness of pollen grains that are dehydrated, and rehydrated, and in the presence of CaCl_2 (a crosslinking trigger) and find significant differences in stiffness both within each sample (between the intine and the exine) and between samples under varying conditions. Secondly, they measure the stiffness of an adult *C. elegans* worm at different locations on the body using the indentation technique concluding that variations in body stiffness along the body arise from differences in internal body structures.

Response:

We thank the reviewer for his/her time and effort spent in reviewing our manuscript. His/her inputs were crucial for improving our work, specifically the suggestions related to statistical evaluations.

Overall, the methods section is well written, and easy to follow. The abstract, however, is a little vague. Suggestions to improve the abstract and some major points of the paper are given below. To my knowledge this is the first demonstration of 3D acoustic manipulation being combined with micro-indentation for mechanical measurements of biological samples. However, I would like to note that neither 3D acoustic manipulation (the authors themselves have published the same protocol - D. Ahmed et al, 2016) nor micro-indentation (for example CFM has been used for pollen tubes, again by B. Nelson's group and others) are novel by themselves and have been leveraged individually in many previous studies as the authors themselves have stated.

Response:

Indeed, acoustic manipulation as well as microindentation have been presented previously, as stated in our manuscript. The combination of these techniques, however, does not only come with new and exciting advantages for a variety of research fields but also with significant challenges, which had to be addressed by redesigning the experimental setup as well as making crucial adjustments to the indentation procedure. This was only possible because the groups involved already have extensive experience with these methods. As an example, the acoustic device relied on numerous design adjustments to ensure proper access to the manipulated specimen using the force sensor, whilst ensuring that the indentation probe does not touch the PDMS. Simultaneously, a stiff substrate necessary for mechanical quantification has to be provided, which was not required for previous manipulation approaches. Furthermore, the PDMS design had to be adjusted to allow for improved temporal stability of the microbubble as well as optimized controllability of the specimen at low angular velocities. Additionally, the influence of acoustic excitation on the force sensor has been investigated to prevent damaging the fragile MEMS structure, whilst avoiding the formation of a microstreaming near the sharp tip of the sensor that would significantly influence the cavitation streaming used for manipulation and thus alter the specimen's orientation.

We would further like to highlight that, although a technique is known, this does not automatically mean that the applications are not novel. Consider the case of atomic force microscopy, which is well established but can still be applied to gain surprising and fascinating insights with substantial importance for a large variety of research fields.

Therefore, we think that through the combination of previously established techniques and with adjustments and optimizations, original work can be created to expand our current understanding and gain novel insights; in this case, through the first 3D quantification of mechanical properties of pollen grains and *C. elegans* nematodes.

Based on the reviewer's comments, we have added a description of some of the challenges associated with the combination of acoustic manipulation and microindentation to alert future readers interested in adapting our approach to potential challenges and to improve the accessibility of our technique. For example, the following section has been adjusted (line 501): "To prevent possible damage to the fragile MEMS force sensor as well as radiation force-based attraction between the sensor probe and the specimen during acoustic excitation, it is crucial to position the force sensor at a distance of at least 20 μm from the sample. However, the exact position might vary depending on the applied input power and the resulting strength of the acoustic field."

Given the potential utility of this tool, I would have thought the authors would leverage it to develop a more cohesive set of results on either one of the systems (i.e. pollen grains or *C. elegans* worms) and push further than being a simple demonstration of combining two well known techniques. On this front, I would rate the novelty relatively low for Nature communications.

Response:

We think that our presentation of the results may have led to some confusion and would like to apologize for this inconvenience. We would like to highlight that the presented dataset consists of 650 measurements on pollen grains. While 50 measurements are used to quantify the mechanical properties of the exine layer of dehydrated pollen grains, 300 measurements are used to quantify the different layers, i.e., the intine and exine, of hydrated pollen grains in deionized water and additional 300 measurements are performed on hydrated pollen grains in CaCl_2 solution to detect changes in mechanical stability associated with the environment. Furthermore, we present 50 data points gained from the indentation of a single *C. elegans* nematode at different orientations of the specimen, which allow the observation of varying stiffness values on a single animal. These datasets demonstrate the importance of our technique for the investigation of small yet complex specimens as the reported variations were previously unknown and common quantification methods lacking manipulation capabilities were unable to differentiate between variations caused by material changes on individual specimens and biological variations between multiple samples. Nevertheless, to address the reviewer's comment, we significantly expanded the investigation on pollen grains by using in-depth numerical simulations to disentangle the complicated relationship between material stiffness and turgor pressure, which both contribute to the experimentally determined apparent stiffness values presented in the previous version of the manuscript. Hereby, we present for the first time the elastic properties of the highly complex bilayered cell wall of the pollen grain and demonstrate that the Young's modulus of the exine is more than 10 x higher than that of the intine. Furthermore, we have extended the experimental dataset by investigating the repeatability of our indentation approach for the mechanical characterization biological specimens ($m = 50$). In addition, the manuscript has been adjusted in numerous places to prevent misunderstandings as well as a mix-up between experiments and datasets.

We are convinced that the changes described above will serve to make the manuscript clearer, drastically increase its novelty factor, and significantly broaden the circle of potentially interested readers.

As indicated below, the data they present already includes some interesting trends that could be investigated further. However, the techniques presented here do offer several advantages for stiffness measurements of biological samples. The fact that the technique is not specimen dependent and that it appears relatively easy to implement allows it to be more general and relevant for a very wide audience.

Response:

We thank the reviewer for acknowledging our simple yet powerful technique that could have a broad impact on a variety of research fields, including biology and biomedical research.

Comments on the Statistics: The addition of CaCl_2 leads to “an intine stiffness which is on average 0.7 times the stiffness value of the exine.” What is the error on that value? Is it really significant as compared to the 0.56 in the previous ‘dry’ experiments? When looking deeper, one finds in the SI that the precise value is 0.66. I have two comments regarding the statistics here:

Response:

We thank the reviewer for this comment as it highlights the importance to further improve our statistical evaluations. Unfortunately, there seems to have been some confusion regarding the various experimental sets with pollen and we apologize for not having described this in a clearer way. The presented data for pollen are based on three sets of experiments rather than only two as mentioned by the reviewer. For the first evaluation of pollen grains in H_2O , i.e., 30 pollen with 300 measurements, the average stiffness ratio is 0.56. The second set contains measurements of pollen in a CaCl_2 solution, again with 300 new measurements on 30 pollen grains, and the average stiffness ratio is 0.66. This difference has been found to be significant with a p-value of 0.000312, as described in the Supplementary Information SI4. Additionally, the apparent stiffness of dry, dehydrated pollen grains has been characterized (on 50 pollen) as 194 N/m. It is important to highlight that, for the case of dehydrated pollen grains, only the exine can be detected and, therefore, no stiffness ratio has been derived. We would further like to apologize for the inconsistency regarding the precise values of the stiffness ratios for pollen grains in CaCl_2 solution. For readability purposes, the previous manuscript contained rounded values, which led to the average apparent stiffness ratio for pollen grains in calcium chloride solution of 0.7.

Per the reviewer’s input, we have made several additional changes associated with the presentation of the statistical evaluation, such as communicating the precise p-value in the main manuscript as well as providing the exact apparent stiffness ratio in the sentence (line 276) “Given the experimental data, it can be seen that CaCl_2 leads to an apparent stiffness for the intine which is on average 0.66 times the apparent stiffness value of the exine.”.

Furthermore, the manuscript has been refined at numerous places to remove such ambiguities and to improve the general readability and accessibility of the results. As an example, the following sentences have been added to the manuscript or adjusted to improve the introduction of the mechanical characterizations we performed (line 74): “We characterize the apparent stiffness ratio based on the apparent stiffness of different cell wall components of individual pollen grains in deionized water and examine our experimental observations through numerical simulations to disentangle the complex, intertwined contributions of geometry, heterogeneous material composition, and turgor pressure on the apparent stiffness obtained from mechanical characterizations. We then proceed by investigating the influence of the environment on the structural stability of the plant specimen through the quantification of pollen grains in water or calcium chloride (CaCl_2) solutions and compare our observations to apparent stiffness values obtained for dehydrated specimens.”

Both intine and exine data look bimodal – there might be something of interest there – I would advise the authors to look more deeply at their data.

Response:

We thank the reviewer for sharing his/her thoughts concerning the apparent stiffness values presented for hydrated pollen grains. Based on his/her suggestion (see below), we expanded our statistical evaluation to include the D'Agostino omnibus K2 normality test. The corresponding results confirm the reviewer's initial observation for the distribution of intine measurements ($K2 = 14.2$, $p = 0.0008$) as well as exine measurements ($K2 = 11.47$, $p = 0.0032$) with both being clearly non-normal. Therefore, we investigated the cause of this behavior in more detail and tried to evaluate a common characteristic of the measurements that would allow us to separate them into two categories following a bimodal distribution. Unfortunately, it was not possible to separate the data based on the used force sensor, the date of the experiment, time of the measurement (e.g. more time to hydrate), the plant source (i.e., flower), or the size and shape of the pollen. As an example, Supplementary Fig. SF2a shows two geometrically similar pollen grains exposing different apparent stiffness values that were quantified on the same date using the same sensor and were taken from the same flower. Therefore, we assume that this bimodal behavior might be caused by the existence of more than one stable configuration of mechanical parameters of the pollen grains. For example, it could reflect the difference between rehydrated yet not fully pressurized and turgid pollen grains. However, further investigations, which are clearly beyond the scope of this manuscript will be required to answer this question.

Based on the reviewer's input, the following statistical evaluation has been added to the supplementary material (line 183 and line 48): "Given the near bimodal nature of the obtained apparent stiffness values for pollen grains, all data has been tested for normality using a D'Agostino omnibus K2 prior to the statistical evaluation through t-tests. Intine apparent stiffness ($K2 = 14.2$, $p = 0.0008$) as well as exine apparent stiffness ($K2 = 11.47$, $p = 0.0032$) from pollen in deionized water were found to show a non-normal distribution. Combined intine as well as exine measurements obtained from lily pollen grains in deionized water have been detected as not normally distributed ($K2 = 17.95$, $p = 0.0001$). Combined intine as well as exine measurements obtained from lily pollen grains in CaCl_2 solution were found to show a non-normal distribution ($K2 = 33.85$, $p < 0.0001$)."
"The pollen grains were characterized on the same date using the same force sensor and were taken from the same flower sample."

Additionally, the evaluation has been highlighted in the manuscript (line 192): "It is worth noting that the apparent stiffness distribution for intine as well as exine measurements was found to be significantly different from normal (see Supplementary Information S13). However, no correlation was found with the force sensor used, the date of the experiment, the time of the measurement, the flower from which the pollen was derived, or the geometry of the pollen grain."

While the data may be statistically significant I am not sure these small differences as really significant. At the very least, using a non-parametric test might be better since we cannot assume a normal distribution (unless authors have verified that the distribution of means is normally distributed).

Response:

We would like to highlight that all statistical evaluations have been performed on the derived stiffness ratios and not on the directly measured apparent stiffness. As mentioned in the response to the previous point of reviewer 2, we expanded our statistical evaluation with a D'Agostino omnibus K2 normality test to ensure proper treatment of our data. The stiffness ratio for pollen grains in deionized water ($K2 = 0.3949$, $p = 0.8208$) as well as pollen grains in CaCl_2 solution ($K2 = 2.189$, $p = 0.3347$) results are non-significant and can, therefore, be

assumed to be normally distributed. Moreover, since the F-test showed that the variances are unequal, we used a two-tailed t-test for independent samples and unequal variances, which revealed that the observed differences between specimens in H₂O (M = 0.56, SD = 0.12) and CaCl₂ solution (M = 0.66, SD = 0.08) are significant with $p = 0.000312$. Given that this is a very small p value, compared to a typical threshold of $p = 0.05$, the reviewer can be reassured that the significance is not limited to the statistical evaluation.

However, the absence of bimodal behavior for the apparent stiffness ratios (k_i/k_e) provides us with further clues to possible causes for the bimodal behavior of the apparent stiffness values, as the stiffness ratios, which are calculated for each pollen individually, do not exhibit this distribution.

The corresponding theory has been added to the manuscript (line 379): “It is worth noting that the bimodal behavior observed for the apparent stiffness of the intine as well as the exine has been eliminated through the derivation of the individual stiffness ratios. While substantial experimental and numerical investigations might be required to determine the cause of this bimodality, we propose that it might arise because pollen with a higher exine stiffness also expose higher intine values to form mechanically stable configurations, and vice versa. However, another possible explanation might be differences in hydration, as unfolded pollen do not have to be fully pressurized and a resulting difference in turgor pressure would simultaneously affect intine and exine measurements.”

Per the reviewer’s suggestions, the statistical evaluation has been expanded and the following sections have been added to the manuscript (line 277) and supplementary material (line 191): “Given the bimodal trend of the apparent stiffness, the stiffness ratios have been tested for normality (see Supplementary Information SI3)...” “Normality tests for apparent stiffness ratios k_i/k_e from lily pollen grains in deionized water ($K2 = 0.3949$, $p = 0.8208$) and pollen grains in CaCl₂ solution ($K2 = 2.189$, $p = 0.3347$) are not significant, i.e., the data can be treated as normally distributed. ... Please note that t-tests have only been applied to stiffness ratios, which do not display bimodality and can be assumed as sampled from a Gaussian distribution.”

I hope the authors can rewrite this section more accurately and include more details.

Response:

Based on the reviewer’s suggestion, we have significantly improved the statistical evaluations and their discussion in the manuscript and the supplementary information, added further investigations concerning the distribution of the experimental results, adjusted the discussion of the results as well as the introduction of the different experiments to prevent misunderstandings.

For the measurement of the dehydrated pollen grain in SF3 – taking the mean of the measurements is not a good representation of the distribution (given that it looks nearly bimodal).

Response:

Based on the reviewer’s input, the stiffness measurements for non-hydrated pollen grains have been analyzed for normality. The corresponding D’Agostino-Pearson “omnibus K2” normality test was not significant ($p = 0.0578$) with a K2 value of 5.703. While we agree with the reviewer that the distribution visually tends towards a bimodal behavior, this is not statistically significant according to a D’Agostino-Pearson test and the representation through a mean and standard deviation is appropriate. Nevertheless, we agree with the reviewer that this behavior requires further discussion and, therefore, highlighted it in the manuscript.

The following sections in the manuscript (line 291) and supplementary material (line 193) have been adjusted to account for this discussion: “Furthermore, it is important to highlight that the data obtained for dehydrated pollen grains follows a nearly non-normal distribution ($p = 0.0578$). Therefore, further investigations would be required ...” “Measurements from folded lily pollen grains have been found normally distributed ($K2 = 5.703$, $p = 0.0578$), albeit the near-significance of the result leaves room for discussion.”

In Fig 3F, could the authors state how many worms were used to determine that the body wall muscle leads to higher stiffness measurements?

Response:

All presented measurements concerning *C. elegans* were performed on a single specimen to prevent influence from discrepancies between samples. The specimen has been re-oriented using the acoustic setup to allow access to the different surface regions. The observed variation is thus not be caused by differences between multiple nematodes. While similar apparent stiffness values have been observed for other specimens, Fig. 3F only contains the results of the single sample with the best coverage to allow for a reliable interpretation of our observation and to prevent noise caused through biological variation. It is important to highlight that our statement that the body wall muscle may cause higher apparent stiffness values is only an assumption based on the spatial correlation of the higher stiffness values and the body wall muscle of these worms. Whilst a large-scale study would be required to fully address other possible effects as well as the influence of the paralysing drug, this lies outside the scope of the current paper.

Per the reviewer’s input, we have made minor adjustments throughout the manuscript, with multiple sections being modified to better reflect this theory as well as improving accessibility of the results and their derivation for the reader (line 337, line 342, line 408, and line 849):

“Based on the arrangement of the body wall muscles in nematodes as well as the approximate 90° separation of the detected varying stiffness values (see Fig. 3d and green areas in Fig. 3f), the observed difference in apparent stiffness might be caused by the body wall muscles.”

“Therefore, the softer regions would be directly above the eggs/embryos inside the gonad (where the dorsal muscles are not present), while stiffer measurements would be recorded in the region of the body wall muscles.”

“Heterogeneity in measured mechanical properties might be the result of the underlying tissues, such as body wall muscles, over which the apparent stiffness was measured.”

“All presented measurements were performed on a single nematode to avoid noise caused by biological variation between specimens.”

Fig SF3 claims $n=300$ for the pollen grain in CaCl_2 in panel b, is this supposed to be $n=30$ as stated in the text and caption?

Response:

We thank the reviewer for noting this and have adjusted the panel accordingly. Throughout the manuscript, m refers to the number of measurements while n stands for the number of specimens in this experimental group. As pointed out by the reviewer, the correct statement would be $m = 300$, i.e., 300 measurements on pollen in CaCl_2 solution. The figure has been updated accordingly. Furthermore, we checked for similar errors and corrected them (e.g., in the paragraph “The observed large value range over the ...” (line 176)).

Other Suggestions: C. Essmann et al. Nat Comm Feb 2020 should be cited

Response:

The references have been updated to better reflect valuable recent works concerning indentation-based mechanical characterization of C. elegans nematodes: i.e. Essmann et al., Nature Communications (2020) 11, Elmi et al., Scientific Reports (2017) 7, Sanzeni et al., eLife (2019) 8.

In the abstract, it would be useful if the authors specifically mention the results obtained from the experiments (for example that measuring stiffness at two different positions on the worm revealed differences).

Response:

We thank the reviewer for this comment. The abstract has been adjusted to highlight that local variations in mechanical properties have been observed and draw the reader's attention to these results (line 32): i.e. "... the analysis of single Lilium longiflorum pollen grains, in combination with numerical simulations, and individual Caenorhabditis elegans nematodes. It reveals local variations in apparent stiffness for single specimens, providing previously inaccessible information and datasets on mechanical properties ..."

The authors make the claim that they "demonstrate the utility of [their] platform for measurements at the organism level by detecting the influence of internal organs on the mechanical properties of surrounding tissue in different regions of individual nematodes." For the sake of precision, I do not think their results support this claim. Rather they have shown differences in mechanical properties at two individual locations on the worm body. Whether this difference really arises from the influence of internal organs or from local variations on the cuticle itself is not clear.

Response:

We thank the reviewer for raising this point. We agree that, based on our dataset, it is not possible to make such an absolute claim and, therefore, have adjusted our statement to better reflect that this is only one of several possible explanations. While this work focuses on the introduction as well as presentation of our novel approach through its application on multiple specimens, i.e. plant cells as well as entire animals, extensive future studies will be necessary to evaluate the precise influence of internal organs onto indentation-based characterizations and determine the precise interaction between the various components.

Per the reviewer's input, the following changes have been made to the manuscript (line 344 and line 410): "However, it is important to note that the observed variations might also be caused by local variations in the cuticle or by complex interactions between various factors. Further investigations will be required to differentiate between these interpretations as well as to quantify the precise impact of drug-induced short-term paralysation onto internal tissues and subsequent microindentations."

"However, other factors, such as potential local variations in the cuticle or a combination of multiple effects, cannot be neglected and further investigations will be required to differentiate between these interpretations."

The authors claim methods such as glue for immobilizing worms is invasive, but there is strong evidence that chemically paralysing a worm is also invasive. As the authors themselves point out it has an effect on the body wall muscle and therefore may alter the mechanical properties from the natural state. It is inaccurate therefore to imply that paralysing the worm is non-invasive.

Response:

*We would like to apologize for our imprecise statement and would like to clarify that we did not intend to define chemical paralysation as being non-invasive. Unfortunately, despite being very desirable, we are not yet aware of a technique that allows for indentation-based characterizations, especially given the fragile nature of MEMS force sensors, without relying on some type fixation or paralysation. As requested, the corresponding section has been adjusted to discuss the influence of irreversible fixations more accurately while the drawback and possible implications of chemical paralysis have been highlighted more prominently (line 303 and line 347): “Such approaches are efficient yet gentle when manipulating *C. elegans* and, despite relying on paralyzing drugs, have been shown to better preserve animal health and viability when compared to conventional, manual techniques that commonly involve destructive physical constraints such as irreversible glue. This aspect is especially crucial as it might limit the viability of the specimens and correspondingly our capabilities for long-term biomedical studies, such as research on neurodegenerative diseases for which the worm is preferably studied throughout its life-cycle.”*

“... as well as to quantify the precise impact of drug-induced short-term paralysation onto internal tissues and subsequent microindentations.”

Furthermore, the Discussion section has been extended with the following sentence to draw attention to the possible yet unknown influence of chemical paralysation on mechanical characterizations (line 412): “Furthermore, it is important to highlight that the detected variations might be enhanced through the application of chemical paralysation and its precise impact has yet to be quantified.”

For readability: It would be useful for authors elaborate on how the acoustically actuated microbubbles are produced in the main text, though some of it is included in the SI.

Response:

We thank the reviewer for pointing out that our explanation of this aspect was not clear enough. The microbubbles are trapped automatically inside the fabricated microcavities due to hydrophilic/hydrophobic interactions when depositing liquids similar to water near the (hydrophobic) PDMS device.

Based on the reviewer’s suggestions, the following sentence has been added to the manuscript to improve accessibility (line 94): “Due to hydrophilic/hydrophobic interactions, the injected liquid causes air to be trapped within the predefined microcavities of the hydrophobic PDMS, leading to the formation of locally confined microbubbles.”

It is not clear how the specimen is delivered to the manipulation device – in a close microfluidic device, flow is used. In this open system, is it simply placed close to the bubbles? Please specify

Response:

*The reviewer is absolutely correct with his/her assumption. The specimen, in the case of pollen grains already submerged in the liquid, is placed near the microbubble. By acoustically exciting the bubble, acoustic radiation forces are generated that act onto the specimen and pull it towards the bubble/liquid interface where it is trapped and rotated through the acoustic streaming. The use of acoustic radiation forces is particularly of interest for larger specimens, such as *C. elegans*, as the force scales positive with the volume of the specimen, leading to stronger interactions and improved trapping capabilities.*

The following sentence has been added to the manuscript (line 108): “In contrast to closed microfluidic devices, the specimens can be directly positioned near the microbubbles, e.g., using pipettes, and no external pumps are required.”

Is the piezo positioning stage used in combination with microscopy to position the force sensor? Please clarify in text.

Response:

The reviewer's assumption is correct. The protocol for the indentation is performed as follows: Initially, the 3D positioner is used to position the force sensor in the microscope's field of view and fixed at this location. Then, the microscope stage is used to position the sample underneath the force sensor, which has the advantage that the sensor's location remains fixed with respect to the microscopic image and allows for precise positioning. The indentation is performed through an additional high-precision z-stage to ensure full control throughout the mechanical characterization.

The manuscript has been updated as follows (line 126 and line 437): "A 3D positioner navigates the sensor probe close to the specimen and positions it in the field of view before fine positioning is performed via the microscope stage that moves the manipulation device and the sample. Vertical indentations of the exterior surface of the specimen is then performed in z-direction using a high-precision piezo stage."

"The force sensor is connected to a 3D positioner ... which is used to position the micro-indenter inside the field of view as well as close to the specimen prior to the experiment. The sensor keeps its relative position with respect to the microscope even when the microscope stage (containing the acoustic manipulation setup with the specimens) is moved. The sub-system connected to the sensor is placed onto a piezo flexure-guided nanomanipulation system ..."

Please specify how long each measurement requires (for methods section)

Response:

A single indentation (movement of the force sensor to indent the specimen) takes between 10 and 15 seconds. However, loading the device with samples, positioning the sensor, as well as manipulating the specimen between the indentations leads to a total of approximately 10 minutes for a full characterization of a single pollen grain. Furthermore, it is worth noting that only experiments with 10 clean measurements, i.e., without noise in the force-distance curve, have been used for the final statistical evaluation. Therefore, only about every second quantified specimen has been considered for the evaluation and included in this submission.

The requested information has been added to the manuscript (line 454): "Once a specimen is loaded into the setup and positioned appropriately, a single indentation takes approximately 10 - 15 seconds, which is mainly defined through the indentation velocity and the approach distance, i.e., the initial distance between the sensor and the specimen prior to being in contact (e.g., 20 μm)."

Pg 9. A clarification is required in the statement: "the less soft regions are directly above the eggs/embryos inside the gonad (where the dorsal muscles are not present), while stiffer measurements were recorded in the region of the body wall muscles". Do they mean soft or less soft? I am assuming less soft means stiff! This might just be a typo.

Response:

We thank the reviewer for detecting this mistake. The source data has been checked once more and the typo, as correctly assumed by the reviewer, has been corrected.

Since this is a methods paper, please include the schematic/photograph of the setup in the main text Fig1 rather than in the SI

Response:

Per the reviewer's input, Fig. 1 as well as the Supplementary Fig. SF1 have been updated to include two different images of the setup that allow the reader to get a better idea of the system and improve accessibility. Furthermore, a technical drawing of the sensor holder as well as the mask design of the acoustic device used for its fabrication have been made available through Github (see Data Availability) and the Materials and Methods section has been improved to provide a link to detailed descriptions of the setup components for interested readers.

Reviewer #4 (Remarks to the Author):

In this article the authors report on the combination of an acoustically driven manipulation device to trap and rotate biological samples and a micro-indenter. The device consists of a piezoelectric transducer mounted on a glass slide together with a PDMS structure. The main principle is given by microbubbles caught in the cavities of this structure. These microbubbles are excited acoustically leading to the well known acoustic streaming. As inertia only plays a minor role, this allows to precisely rotating the trapped object as desired. The structure of the article is clear and all figures are very carefully and nicely prepared. The methods seem solid to the widest extent (minor suggestions see below).

Response:

We thank the reviewer for assessing our manuscript and are pleased that the effort put into the presentation of our results has been acknowledged.

However, I find the article is not suited for publication in nature communications due to the following reasons: The study is a basic combination of earlier work of the group (force microscopy+acoustic rotation).

Force Microscopy: Vogler et al. / The Plant Journal(2013)73,617–627, Shamsudhin et al. / PLOS ONE | DOI:10.1371/journal.pone.0168138 December 15, 2016, Burri et al. / IEEE ROBOTICS AND AUTOMATION LETTERS, VOL. 4, NO. 2, APRIL 2019. Acoustically induced rotation: Nino Läubli et al. / Procedia CIRP 65 (2017) 93 – 98, Nino F. Läubli et al. / Small Methods 2019, 3, 1800527. Thus the advance for me seems too incremental to be suited for Nature communications.

Response:

It is correct that, separately, acoustic and microindentation have previously been presented. However, the combination of different techniques and interdisciplinary approaches are crucial to allow us to gain new insights into a large variety of research related to biology or medicine. Whilst we have used both methods previously, we are sure that the reviewer is aware of the numerous challenges that arise through the combination of different techniques and that their optimization significantly relies on experience as well as previously developed expertise. Thus, we would like to highlight some of the challenges and additional requirements of this work that went way beyond a “basic” combination and required further investigations and redesigns. While acoustic manipulation in open microchannels has been shown previously, the current setup uses a different geometrical design for the PDMS device, which also led to a significantly different streaming pattern, mostly consisting of out-of-plane vortices for all investigated excitation frequencies (as present in Supplementary Fig. SF2 b). Additional requirements for the PDMS device include, in contrast to the open channel arrangement of the previous publication used for sole manipulation purposes, its combination with a stiff substrate to allow for reliable mechanical characterizations as well as proper accessibility with the force sensor in z direction (while the tungsten probe diameter varies along the height from 2 μm to up to 35 μm), or improvement of the temporal stability of the microbubbles through optimizations of the microcavities. Additionally, one must consider the vibrations of the nearby force sensor during re-orientation of the samples, which requires improved controllability over the acoustic device to prevent damage to the MEMS structure. Furthermore, such vibrations can also lead to streaming formations near the sensor as well as attraction forces between the sensor and the specimen (due to the radiation forces near sharp tips) but also critically damage the fragile force sensor. To ensure that such issues can be prevented by those adapting our approach, we spent much effort demonstrating our method and characterization procedure, and prepared the manuscript accordingly. We have further improved accessibility and knowledge transfer through in this revision process.

The following parts has been added to the manuscript to reflect some challenges associated with acoustic excitation and fragile MEMS sensors as well as to highlight our observations concerning the shape of the acoustic streaming (line 372 and line 511): "Additionally, high controllability over the produced microstreaming is essential to prevent damage to the fragile MEMS force sensor by inducing vibrations and to avoid attractive radiation forces between its sharp tip and the investigated specimen, which could alter the sample's orientation." "... because it allows us to expose parts of the specimen that were previously obstructed as well as due to its strong presence throughout all investigated excitation frequencies."

Additionally, it somehow irritates me, that the original work of the group presenting the rotation method (Nino F. Läubli et al. / Small Methods 2019, 3, 1800527) is very close to work published already by one of the authors in 2016 in NC (Ahmed et al. NATURE COMMUNICATIONS | 7:11085 | DOI: 10.1038/ncomms11085).

Response:

We take note of the reviewer's comment. While the publications mentioned indeed both refer to sample rotations and are thus necessary for the realization of our objective, they are only loosely related to the 3D mechanical characterization of the present manuscript. Nevertheless, we would like to point out that there are significant differences between the methods and applications described in the publications mentioned by the reviewer. These include the analysis of open microchannel structures, use of a single bubble geometry for all manipulations, investigation of the influence of the rough surface of non-spherical plant material onto our manipulation capabilities, as well as adaptation for 3D high resolution reconstruction through non-calibrated microscopes. The last part is especially of interest for various research groups and the accomplishment highlights the simplicity and adaptability of our acoustic manipulation method.

Many claims (e.g. "Acoustofluidic devices generally consist of closed microchannels") are probably not really justified, which might be caused by a lack of knowledge of existing literature from the early 2000s. This lack also affects the reference list, where some fundamental pioneering papers are missing.

Response:

We thank the reviewer for sharing his/her expertise of the field of acoustofluidics and agree that more emphasis could have been put onto the acknowledgment of previous work concerning acoustically excited bubble-based manipulation. Unfortunately, the initial submission has been internally transferred from another journal with stronger limitations on the permitted number of references. Given the strong interdisciplinary nature of our manuscript combining four research fields, i.e. acoustics, mechanical characterization, pollen grains, and *C. elegans*, a large number has already been occupied through the corresponding introductions. Therefore, we would like to apologize for any inconvenience caused.

Per the reviewer's suggestion, the corresponding sentence has been adjusted (line 311): "As, unfortunately, many acoustofluidic devices rely on closed microchannels to combine acoustic excitation with hydrodynamic flows, direct access to the constrained animals is strongly limited."

Furthermore, to highlight important and ongoing research on acoustically excited microbubbles, the following references have been added to the bibliography: Marmottant and Hilgenfeldt, *Nature* (2003) **423**, Marmottant and Hilgenfeldt, *Proc. Natl. Acad. Sci* (2004) **101**, Combriat et al., *Soft Matter* (2020) **16**, Jiang et al., *Lab Chip* (2021), issue not yet available. To highlight pioneering and recent work in open as well as closed setups we also included the sentence (line 60): "...acoustofluidic excitation has been used for particle and

sample manipulation and rotation via surface acoustic waves, solid actuators, or trapped microbubbles in closed fluid chambers as well as open setups.”

Nevertheless, the manuscript seems publishable in another journal claiming less ground breaking advance with only minor modifications.

Response:

*We hope that through our extensive revision of the manuscript, including clarification of the challenges of our approach, and by addressing the reviewer’s concerns in this response, we have been able to further highlight the importance of our work and to convince reviewer 4 to reconsider the possibility of publishing our manuscript in Nature Communications. We think that the combination of two known techniques engenders substantial novelty and that such systems can be of great importance for a variety of research fields, including biology and biomedical research. As an example of novel insights that have only been possible through the demonstrated method, we were able to observe and quantify the stiffness variations on individual *C. elegans* worms. Although the model organism is of broad interest and indentation as well as acoustofluidic manipulation has been known for a long time, only through their combination such novel investigations and insights have been possible.*

For this purpose, I would recommend to apply a model (Hertz?) for the indentation measurements to present quantitative values of the Young’s modulus.

Response:

We would like to thank the reviewer for the suggestion to apply a modelling approach to assess the correlation between the various mechanical properties of the specimen affecting the experimentally determined apparent stiffness. While the use of a Hertz model would be insufficient to account for such complex, biological specimens, including their variation in material composition, we opted for an in-depth numerical simulation of an average pollen grain to shed light onto the relationship between apparent stiffness, material stiffness, i.e., Young’s modulus, and internal turgor pressure.

The corresponding results are presented in a newly added section Numerical Simulation of Plant Cell Indentation (line 198) as well as discussed in Materials and Methods (line 537), Discussion (line 387), and Supplementary Material through the means of Figures (Supplementary Fig. SF 4), Tables (Supplementary Table ST1), and Supplementary Information (Supplementary Information SI6, line 208).

The insights gained through our investigation of this matter have been directly incorporated into the manuscript and led to extensive adjustments and corrections, such as the distinction between apparent stiffness and material stiffness. Furthermore, the detailed numerical study allowed us to highlight the complexity of mechanical quantifications and the relevance of our approach to allow for the quantification of multiple parameters on single specimens and, by that, reduce the noise caused by biological variation.

Moreover, I would recommend to supply some additional statistical test (t-test or other suitable) to additionally support the biological findings.

Response:

*We thank the reviewer for this comment. We would like to highlight that the statistical evaluation of our experimental results through t-tests, in combination with F-tests, was already provided in the previous submission as part of the Supplementary Information (Supplementary Information SI4 for pollen grains and SI5 for *C. elegans* nematodes).*

However, the statistical evaluation has been further improved and expanded through D’Agostino omnibus K2 normality tests (Supplementary Information SI3, line 182) and the

corresponding observations have been added to the manuscript for discussion. Furthermore, we experimentally quantified the repeatability of our mechanical characterization method for biological specimens (Supplementary Fig. SF1b) and performed the corresponding statistical evaluations (Supplementary Information SI2, line 168).

Additionally, based on the reviewer's input, the manuscript has been improved to ensure that future readers are aware of the performed statistical evaluations, e.g., on line 279 and line 336: "... the difference of the stiffness ratios obtained for pollen in deionized water and CaCl_2 solution has been found significant ($p = 0.000312$) in subsequent statistical evaluations (see Supplementary Information SI4)"
"... were found to be significantly different ($p = 3.14e-10$) through statistical evaluations (see Supplementary Information SI5)".

In summary, I do not recommend publication of this work in Nature Communications for the reasons given above. It does not meet the criteria.

Response:

We take knowledge of the reviewer's concerns regarding the novelty of our manuscript but hope that through the significant revision of our manuscript as well as our direct responses to the criticisms raised, we were able to highlight the importance and relevance of our work for a large variety of research applications.

Reviewers' Comments:

Reviewer #2:

Remarks to the Author:

Authors have carefully addressed the comments by reviewers. The addition of the numerical simulations has also added to the article. The question of novelty lingers, but I think the the fact that the technique has potential to be useful for many applications warrants publication.

Reviewer #4:

Remarks to the Author:

The authors significantly strengthened the manuscript with this revision. Moreover, it might be indeed more complicated to combine two existing approaches as argued by the authors. In addition, the application of the technique to plant experiments seems of high interest for the according community. Taken together and considering the demonstrated attention to the detail, I can recommend to accept the manuscript for publication.

Reviewer #5:

Remarks to the Author:

I thank the authors for addressing the concerns of reviewer 1 to compare their measurements to the previous publications. However, the data for stiffness measurements have enormous variation. It is disappointing that they have not been able to determine the reason for the variation or the bimodal distribution, a point also raised by another reviewer. The field of plant biomechanics is suffering due to a large amount of data that contains large variation and the lack of reproducibility between samples.

As a minor comment

I would request that they clarify the sentence "measurements was found to be significantly different from normal" that they refer to the normal distribution rather than normal in the sense of usual to avoid confusion.

Point-by-Point Response to the Reviewer's Comments

First, we would like to thank the reviewers for the evaluation of our revised manuscript and providing their constructive feedback. Below you can find a point-by-point response to the reviewers' comments. For ease of reading, the reviewers' comments are shown in blue and our responses in italics.

Reviewer #2 (Remarks to the Author):

Authors have carefully addressed the comments by reviewers. The addition of the numerical simulations has also added to the article. The question of novelty lingers, but I think the fact that the technique has potential to be useful for many applications warrants publication.

Response:

We thank reviewer 2 once more for his/her valuable inputs. Specifically his/her suggestions regarding the statistical evaluation significantly improve the quality of our evaluations and we are glad to hear that we were able to address his/her critics through the revision of our manuscript.

Reviewer #4 (Remarks to the Author):

The authors significantly strengthened the manuscript with this revision. Moreover, it might be indeed more complicated to combine two existing approaches as argued by the authors. In addition, the application of the technique to plant experiments seems of high interest for the according community. Taken together and considering the demonstrated attention to the detail, I can recommend to accept the manuscript for publication.

Response:

We acknowledge the reviewer's response and are pleased that, through the revision of our manuscript, we have been able to address his/her concerns regarding the novelty and challenges associated with our approach.

Reviewer #5 (Remarks to the Author):

I thank the authors for addressing the concerns of reviewer 1 to compare their measurements to the previous publications. However, the data for stiffness measurements have enormous variation. It is disappointing that they have not been able to determine the reason for the variation or the bimodal distribution, a point also raised by another reviewer. The field of plant biomechanics is suffering due to a large amount of data that contains large variation and the lack of reproducibility between samples.

Response:

We thank the reviewer for evaluating our revised manuscript with a focus on reviewer 1's inputs. While, unfortunately, we were not able to determine the cause of the observed stiffness variations, we agree this is a problem not only limited to our work but mechanical characterizations of plant specimens in general. We would like to emphasize that with our

approach we could at least reduce the variations due to different specimen orientations. However, given the highly complex yet often oversimplified morphology of small organisms, we believe that future research has to rely on substantial collaborations between various disciplines including biology, chemistry, and engineering, to successfully unravel the complex relation between the numerous parameters affecting each individual measurement.

As a minor comment

I would request that they clarify the sentence "measurements was found to be significantly different from normal" that they refer to the normal distribution rather than normal in the sense of usual to avoid confusion.

Response:

We thank the reviewer for this valuable input to further improve the accessibility of our manuscript and prevent possible confusions. Per the reviewer's suggestion, the sentence has been adjusted (line 192):

"It is worth noting that the apparent stiffness distributions for intine as well as exine measurements were found to be significantly different from a normal distribution."